# Genome-wide association analyses of autoimmune hypothyroidism reveal autoimmune and thyroid-specific contributions and an inverse relationship with cancer risk

Mary Pat Reeve [1,2] ✉, Masahiro Kanai [2,3,4], Daniel B. Graham [2,4], Juha Karjalainen[1,2], Shuang Luo [1], Nikita Kolosov [1,5,6], Cameron Adams[7], Jarmo Ritari [8], Konrad J. Karczewski [2,3,9], Tuomo Kiiskinen [10], Yu Jiang[2], Zachary Fuller [2], Juha Mehtonen [1], Mitja I. Kurki[1,2,3], Zia Khan[7], FinnGen*, Jukka Partanen [8], Mark I. McCarthy[7], Mykyta Artomov [1,5,6], Aarno Palotie [1,2,3], Tiinamaija Tuomi [1,11,12,13], Matti Pirinen [1,14,15], Jukka Kero [16,17], Ramnik J. Xavier[2,4], Mark J. Daly [1,2,3] & Samuli Ripatti [1,2,3,18]

The high prevalence (>5%) of autoimmune hypothyroidism (AIHT) provides a unique opportunity to dissect genetic contributions to systemic and organ-specific autoimmunity. Here we performed a genome-wide association meta-analysis of 81,718 AIHT cases in FinnGen and the UK Biobank, identifying 418 independent signals ($P < 5 \times 10^{-8}$). At 48 of these loci, a protein-coding variant is, or is highly correlated ($r^2 > 0.95$) with, the lead variant, including Finnish-enriched coding variants in *LAG3*, *ZAP70* and *TG*. We demonstrated that *ZAP70*:T155M reduces T cell activation and broadly compare large-scale scans of nonthyroid autoimmunity and thyroid-stimulating hormone levels with a Bayesian classifier to assign loci into distinct groupings, estimating that 38% are involved in general autoimmunity whereas 20% are thyroid specific. We further identified substantial antagonistic pleiotropy, with 10% of AIHT loci showing a consistent protective effect against skin cancer. The AIHT results, including numerous genes encoding checkpoint proteins, support the causal role of natural immune variation influencing cancer outcomes.

Hypothyroidism is estimated to affect at least 5% of individuals, although underdiagnosis suggests that this may be a substantial underestimate[1,2]. In areas of the world with iodine sufficiency, the most common cause of hypothyroidism is Hashimoto's disease, an autoimmune attack on the thyroid leading to reduced thyroid hormone production. As such, it constitutes the most common autoimmune disease[3], although at the same time it is considered vastly underdiagnosed considering the nonspecific and gradual onset of

the many clinical symptoms that it causes. Early detection is key because treatment and supplementation with levothyroxine (synthetic thyroid hormone ($T_4$)) can largely alleviate symptoms and prevent longer-term complications.

Given the well-established sharing of genetic risk factors across autoimmune diseases, a genetic study of hypothyroidism at scale would be expected to provide general insights into autoimmunity, as well as specific insights into thyroid disease, which might aid in early

detection and effective treatment before substantial thyroid damage has taken place.

Population-wide biobank resources enable large-scale integration of clinical information across medical domains. Here we utilize FinnGen's integration of genome information with lifelong medical history in 10% of the Finnish population to detect all treated cases of hypothyroidism (removing major non-Hashimoto's causes of hypothyroidism) and perform a genome-wide association study (GWAS) of AIHT. Repeating the same definitions in the UK Biobank, we present here a genome-wide analysis of >80,000 AIHT cases identifying 418 independent associations, far beyond recent studies[4-7].

## Results

### Phenotypic definitions from registry data

To pursue a GWAS of AIHT first required phenotypic definitions that capture a large but specific set of AIHT. Unlike most autoimmune diseases diagnosed in specialty clinics with well-recorded data, many AIHT cases are detected in primary care and recognized through continuous levothyroxine use. Use of such data required careful removal of individuals who were hypothyroid due to thyroid ablation from Graves' disease, thyroid cancer, thyroidectomy and congenital thyroid disease (Methods). Removal of nearly 10,000 such individuals from the broadest definition resulted in 54,752 cases of AIHT in FinnGen R12, yielding 231 genome-wide significant loci. Confirming our hypothesis that phenotypic restrictions would create a more homogeneous genetic phenotype, the larger, less-specific GWAS of 64,082 individuals treated for hypothyroidism contained substantially fewer (204 versus 231) genome-wide significant associations. Of note, we hypothesized that, although Graves' disease and Hashimoto's disease likely share autoimmune components, there might also be variants with opposite effects on thyroid function. Thus, we intentionally focused the primary scan on AIHT and describe the alignment of the results with the analogous meta-analysis of Graves' disease.

### Meta-analysis of autoimmune hypothyroidism

Having optimized the phenotypic definition in FinnGen, we implemented the analogous phenotype in UK Biobank (Methods), identified 26,966 cases and ran a standard inverse-variance weighted, fixed-effects meta-analysis. Using a strict linkage disequilibrium (LD)-based definition of independence (Methods), this meta-analysis (81,718 cases, 732,951 controls) produced a total of 417 independent genetic associations outside the major histocompatibility complex (MHC; Supplementary Table 1) and numerous highly significant MHC associations centered at previously reported common variants spanning the *DRB1–DQA1–DQB1* locus[5]. Even conservatively ascribing associations within 1 Mb to the same 'locus' indicates at least 280 distinct genomic regions associated with AIHT. Both increased sample size and improved phenotype specificity led to improved power in this study compared to prior studies (Supplementary Table 10). Replication in the recently released MVP dataset[8] (Supplementary Table 1) shows significant replication (one-sided $P < 0.05$) and directional consistency at 89% and 98% of loci, respectively.

### Overview of association signals

At 48 of 417 associations, the lead variant is itself, or in very high LD (squared correlation coefficient, $r^2 > 0.95$) with, a protein-coding variant (Table 1). Among these are well-established common variants (for example, *PTPN22*, *SH2B3* and *FUT2*), as well as lower-frequency hypomorphic alleles (for example, *TYK2* and *IFIH1*) associated with many autoimmune diseases. Of the associations, 16 are low-frequency variants highly enriched in Finns (from 4-fold to >100-fold; Table 2), 12 of which are found only in the FinnGen GWAS because of their low frequency in the UK Biobank (UKBB). Notably, 6 of these 12 map on to coding variants noted above.

Some 51 lead variants have a minor allele frequency (MAF) <5% in Finland, 16 of which (31.4%) are in the most likely 'coding association'

($r^2 > 0.95$; Table 1) category, whereas only 32 of the 366 (8.7%) more common variants are—a significant ($P < 0.0005$) 3.5-fold excess reflecting both natural selection seen broadly across GWASs (higher effect alleles are kept lower in frequency, although not directionally with respect to disease) and function (higher effect alleles detected in frequency-agnostic GWAS analysis are more often coding than lower effect ones)[9,10].

Lower frequency-associated coding variants provide direct clues to disease biology. Noteworthy findings include a missense variant in *LAG3* (P67T), an inhibitory immune receptor with no prior reported genetic associations for which inhibitors have been recently approved as immunotherapy in advanced melanoma[11,12]. The most notable new Finnish association, however, is a noncoding variant (Chr. 16:27384341:C:CT) ~20 kb from the transcription start site of *IL21R*. In addition, rare missense variants in both *IRF3* and *IRF4* confer protection from AIHT (with effects exceeding protective hypomorphic missense variants at *TYK2* and *IFIH1*), as does a low-frequency risk variant of *NFKBIZ*.

We also observed unexpected pleiotropy. A missense variant in *ZNF800* (P103S) that increases hypothyroid risk also increases cataract risk in FinnGen and lowers alkaline phosphatase levels and bone mineral density in UKBB (all $P < 1 \times 10^{-20}$). Two *PER3* variants in complete LD (Pro415Ala and His417Arg) that lower AIHT risk are associated with morning chronotype in UKBB[13] and reported[14] to segregate in a family with familial advanced sleep phase syndrome and demonstrated to reduce PER3 protein levels and repressor activity.

### Hypomorphic mutation of *ZAP70* drives autoimmunity and immune deficiency

Another new Finnish-enriched association is a missense variant in *ZAP70* (Thr155Met), which encodes a tyrosine kinase essential for signal transduction downstream of the T cell receptor (TCR) in response to antigen recognition. *ZAP70* mutations cause a severe autosomal recessive combined immunodeficiency[15-17] marked by absence of CD8+ T cells and CD4+ T cells that do not respond to TCR-mediated activation. Given prior suggestions that ZAP70 inhibition might be therapeutically efficacious in autoimmunity[18,19], and the parallel to *TYK2*, where an allelic series of immunodeficiency and autoimmune protective alleles led to a recently approved therapeutic, we sought to further elucidate the function of *ZAP70*:Thr155Met.

ZAP70 exists in an autoinhibited conformation at baseline and, on productive TCR engagement, the two tandem SH2 domains of ZAP70 bind to phosphotyrosine motifs in CD3ζ associated with the TCR. This recruitment coincides with phosphorylation of ZAP70 in the interdomain B region and kinase domain by LCK, which elicits full activation of ZAP70, promoting phosphorylation of key substrates such as SLP76 and LAT that propagate the TCR signaling cascade[20].

Several rare variants in *ZAP70* cause Mendelian immunopathologies by ablating protein expression, impairing kinase activity or relieving autoinhibition[21]. Clinical manifestations are heterogeneous, most commonly combined immunodeficiencies with recurrent infections, although many patients exhibit paradoxical autoimmunity and lymphoproliferative syndromes[21]. We sought to determine how the Thr155Met variant impacts ZAP70 function and whether it is associated with a gain of function through loss of autoinhibition or impaired function. Toward this end, we reconstituted ZAP70-deficient Jurkat T cells with variants of interest and monitored TCR signaling. Wild-type, ZAP70-reconstituted cells upregulated expression of the activation marker CD69 and induced phosphorylation of SLP76 and ZAP70 after TCR stimulation, whereas parental ZAP70-deficient T cells did not (Fig. 1). Cells expressing a ZAP70 double tyrosine mutant (Tyr315Ala&Tyr319Ala) within interdomain B, which is incapable of activation, showed a complete block of activation and SLP76 phosphorylation after TCR stimulation (Fig. 1). Cells reconstituted with *ZAP70*:Thr155Met exhibited a partial block in activation and phosphorylation of SLP76 (Fig. 1). Thus, *ZAP70*:T155M associated with autoimmunity impairs TCR signaling strength through an incomplete loss of

**Table 1 | Coding variants implicated by AIHT genome-wide association meta-analyses**

| Lead variant | P value | Allele frequency (FIN) | $r^2$ coding | | Annotation |
|---|---|---|---|---|---|
| 12:111446804:T:C | $9.07 \times 10^{-234}$ | 0.595 | | 1.00 | SH2B3:W262R |
| 2:162267541:C:T | $2.09 \times 10^{-32}$ | 0.585 | | 1.00 | IFIH1:A946T |
| 16:50301163:C:A | $4.66 \times 10^{-30}$ | 0.00872 | | 1.00 | ADCY7:D439E |
| 2:162268127:T:C | $6.91 \times 10^{-28}$ | 0.0194 | | 1.00 | IFIH1:I923V |
| 19:10352442:G:C | $1.17 \times 10^{-24}$ | 0.0309 | | 1.00 | TYK2:P1104A |
| 8:132887335:C:T | $2.01 \times 10^{-21}$ | 0.00145 | | 1.00 | TG:Q655X |
| 1:2562891:G:A | $6.82 \times 10^{-21}$ | 0.121 | | 1.00 | TNFRSF14:V241I |
| 6:166929653:G:A | $6.25 \times 10^{-16}$ | 0.0579 | | 1.00 | RNASET2:R236W |
| 19:11416089:T:G | $9.88 \times 10^{-16}$ | 0.576 | | 1.00 | RGL3:H162P |
| 11:94179472:A:G | $3.41 \times 10^{-14}$ | 0.0859 | | 1.00 | PANX1:I272V |
| 19:48703417:G:A | $1.07 \times 10^{-13}$ | 0.374 | | 1.00 | FUT2:W154X |
| 2:97658891:G:A | $4.22 \times 10^{-13}$ | 0.234 | | 1.00 | ACTR1B:A143V, ANKRD36:K5S_fsTer29 |
| 3:58197909:G:A | $1.70 \times 10^{-11}$ | 0.061 | | 1.00 | DNASE1L3:R206C |
| 19:3179519:C:T | $1.12 \times 10^{-10}$ | 0.0546 | | 1.00 | S1PR4:R243C |
| 19:49665663:G:A | $2.77 \times 10^{-10}$ | 0.00313 | | 1.00 | IRF3:A277T |
| 11:308290:T:C | $4.05 \times 10^{-10}$ | 0.482 | | 1.00 | IFITM2:V33A |
| 13:42574410:C:G | $9.51 \times 10^{-10}$ | 0.0413 | | 1.00 | TNFSF11:P36A |
| 1:185182538:G:A | $1.03 \times 10^{-9}$ | 0.345 | | 1.00 | SWT1:H536R, I148V |
| 10:113588287:G:A | $1.27 \times 10^{-9}$ | 0.0272 | | 1.00 | HABP2:G534E |
| 23:154018741:A:G | $1.60 \times 10^{-9}$ | 0.853 | | 1.00 | IRAK1:F196S, S532L |
| 12:6773332:C:A | $1.67 \times 10^{-9}$ | 0.00098 | | 1.00 | LAG3:P67T |
| 19:21537632:CA:C | $5.24 \times 10^{-9}$ | 0.00724 | | 1.00 | ZNF429:H527L_fsTer157 |
| 7:127375029:G:A | $5.53 \times 10^{-9}$ | 0.0423 | | 1.00 | ZNF800:P103S |
| 3:101852100:G:C | $5.67 \times 10^{-9}$ | 0.0146 | | 1.00 | NFKBIZ:G102A |
| 7:44977336:G:A | $8.71 \times 10^{-9}$ | 0.336 | | 1.00 | MYO1G:V49M |
| 1:113834946:A:G | $<1 \times 10^{-300}$ | 0.855 | | 1.00 | PTPN22:W620R |
| 1:116769497:A:G | $8.58 \times 10^{-9}$ | 0.188 | | 1.00 | CD2:H266Q |
| 10:122389836:C:T | $1.95 \times 10^{-16}$ | 0.29 | | 1.00 | PLEKHA1:T320A |
| 17:39787478:C:A | $1.61 \times 10^{-11}$ | 0.0423 | | 1.00 | IKZF3:G234A, GSDMB:N250G |
| 2:97890066:T:C | $7.60 \times 10^{-24}$ | 0.0193 | | 0.99 | ZAP70:T155M |
| 8:60473974:T:C | $2.30 \times 10^{-18}$ | 0.405 | | 0.99 | RAB2A:L68P* |
| 4:105282863:A:C | $1.08 \times 10^{-8}$ | 0.34 | | 0.99 | TET2:I1762V |
| 3:12209924:G:A | $8.30 \times 10^{-38}$ | 0.771 | | 0.99 | SYN2:T506A |
| 5:139474618:A:C | $3.11 \times 10^{-13}$ | 0.702 | | 0.99 | SMIM33:G88S |
| 22:31146151:A:G | $1.67 \times 10^{-9}$ | 0.782 | | 0.99 | PLA2G3:S70A |
| 22:37187692:G:GC | $1.32 \times 10^{-57}$ | 0.399 | | 0.98 | C1QTNF6:G21V |
| 23:79185670:A:C | $3.53 \times 10^{-26}$ | 0.629 | | 0.98 | GPR174:S162P |
| 16:67628912:T:A | $1.18 \times 10^{-13}$ | 0.039 | | 0.98 | AGRP:A67T, LRRC36:S744G R222P G509S, KCTD19:E750K |
| 17:46019187:G:A | $5.63 \times 10^{-10}$ | 0.081 | | 0.98 | SPPL2C:R461P P643R G620R S224P I471V S601P, MAPT:V364A P277L R455W, KANSL1:I1085T |
| 1:7884770:C:A | $1.87 \times 10^{-8}$ | 0.0269 | | 0.98 | PER3:P415A H417R |
| 22:29579458:A:G | $8.35 \times 10^{-10}$ | 0.804 | | 0.97 | THOC5:V579I |
| 14:105759223:C:T | $1.37 \times 10^{-16}$ | 0.39 | | 0.97 | IGHG3:P221L S314N Y226F, IGHG1:L97R D239E L241M |
| 1:156825994:A:G | $1.44 \times 10^{-24}$ | 0.627 | | 0.97 | SH2D2A:N52S |
| 2:55631052:T:G | $1.35 \times 10^{-9}$ | 0.0367 | | 0.97 | CFAP36:I246F, PNPT1:N590D |
| 11:61128363:A:G | $9.58 \times 10^{-9}$ | 0.612 | | 0.96 | CD5:A471V |
| 18:69860373:A:G | $2.34 \times 10^{-20}$ | 0.536 | | 0.95 | CD226:S307G |
| 9:125354538:A:G | $6.30 \times 10^{-10}$ | 0.508 | | 0.95 | GAPVD1:V334L* |
| 17:7826527:C:A | $2.64 \times 10^{-8}$ | 0.0934 | | 0.95 | KDM6B:T761T762del |

**Table 1 (Continuied)| Coding variants implicated by AIHT genome-wide association meta-analyses**

| Lead variant | P value | Allele frequency (FIN) | $r^2$ coding | | Annotation |
|---|---|---|---|---|---|
| 19:40278113:C:T | $1.54×10^{-15}$ | 0.0106 | | 0.88 | ZNF780B:K284N |
| 17:42137394:T:C | $2.17×10^{-31}$ | 0.203 | | 0.85 | HSPB9:Q2P |
| 11:64340005:T:C | $1.24×10^{-13}$ | 0.385 | | 0.85 | CCDC88B:D193E |
| 6:108989516:T:G | $1.66×10^{-15}$ | 0.824 | | 0.83 | SESN1:L103I |
| 8:132905343:G:A | $9.93×10^{-29}$ | 0.65 | | 0.81 | TG:D1312G |
| 4:186086720:A:G | $1.59×10^{-35}$ | 0.368 | | 0.80 | TLR3:L412F |
| 6:368019:T:A | $6.42×10^{-11}$ | 0.00431 | | 0.78 | IRF4:P88Q |
| 22:41428193:A:AAAT | $6.01×10^{-12}$ | 0.751 | | 0.78 | C22orf46 (noncoding transcript) |
| 5:157186630:T:C | $1.02×10^{-11}$ | 0.106 | | 0.78 | FAM71B:M564T |
| 6:108941239:G:C | $2.55×10^{-9}$ | 0.426 | | 0.78 | ARMC2 (splice acceptor)* |
| 9:124267351:A:G | $9.95×10^{-46}$ | 0.402 | | 0.77 | PSMB7:V39A |
| 1:1238231:G:A | $1.38×10^{-9}$ | 0.154 | | 0.75 | C1QTNF12:C231R A180V |
| 12:56627222:T:C | $5.73×10^{-14}$ | 0.317 | | 0.74 | NACA:V336E |
| 20:33085232:A:T | $1.27×10^{-9}$ | 0.0652 | | 0.74 | BPIFB1:T464S |
| 2:200872011:T:C | $2.17×10^{-17}$ | 0.236 | | 0.74 | NIF3L1:T324I |
| 5:35837132:G:A | $1.51×10^{-21}$ | 0.396 | | 0.74 | IL7R:T244I |
| 3:36979158:T:C | $1.56×10^{-12}$ | 0.371 | | 0.72 | MLH1:I219V |
| 11:10509469:T:C | $1.04×10^{-13}$ | 0.0313 | | 0.72 | IRAG1:A79T |
| 22:38700597:T:C | $1.43×10^{-13}$ | 0.399 | | 0.70 | KDELR3:V199G |

Genome coordinates for lead variants given in human genome assembly GRCh38 (hg38). P values are from the FinnGen–UKBB inverse-variance weighted meta-analysis (Methods) and are uncorrected for multiple testing. Allele frequency (FIN), nonreference allele frequency in the FinnGen cohort; $r^2$ coding, $r^2$ (calculated from the FinnGen imputation reference panel) between lead variant and first coding variant listed. All coding variant annotations (in Annotation column) use MANE select transcript as reported in gnomAD v4, except those marked with an asterisk (**), which pertain to an alternative transcript(s) only.

**Table 2 | Finnish-enriched variants in AIHT genome-wide association meta-analyses**

| Lead variant | P value | Allele frequency (FIN) | Enrichment (FIN) | $\beta$ | s.e. | Coding association |
|---|---|---|---|---|---|---|
| 12:6773332:C:A | $1.67×10^{-9}$ | 0.00098 | inf | 0.63 | 0.104 | LAG3 |
| 8:132887335:C:T | $2.01×10^{-21}$ | 0.00145 | inf | 0.829 | 0.0872 | TG |
| 23:155548829:G:A | $2.36×10^{-8}$ | 0.0265 | 120.2 | −0.113 | 0.0202 | |
| 19:40278113:C:T | $1.54×10^{-15}$ | 0.0106 | 93.6 | 0.252 | 0.0316 | ZNF780B |
| 1:51075821:C:CA | $4.74×10^{-10}$ | 0.0568 | 53 | 0.0901 | 0.0145 | |
| 16:27384341:C:CT | $2.17×10^{-34}$ | 0.0582 | 50.6 | 0.173 | 0.0141 | |
| 2:97890066:T:C | $7.60×10^{-24}$ | 0.0193 | 38 | 0.237 | 0.0235 | ZAP70 |
| 3:152243543:C:T | $4.51×10^{-8}$ | 0.0212 | 37.3 | 0.127 | 0.0233 | |
| 12:519821:G:C | $1.63×10^{-8}$ | 0.0351 | 16.9 | 0.104 | 0.0185 | |
| 6:368019:T:A | $6.42×10^{-11}$ | 0.00431 | 13.3 | −0.376 | 0.0575 | IRF4 |
| 2:100705638:A:G | $3.96×10^{-9}$ | 0.0209 | 10.4 | 0.135 | 0.0232* | |
| 9:3857583:G:A | $5.68×10^{-10}$ | 0.00126 | 9.4 | 0.545 | 0.0879 | |
| 12:56182198:T:G | $6.88×10^{-12}$ | 0.00521 | 7.3 | 0.31 | 0.0452 | |
| 19:3179519:C:T | $1.12×10^{-10}$ | 0.0546 | 5.8 | −0.0888 | 0.0155* | S1PR4 |

Index associations (lead variant) to variants with fivefold or greater frequency in Finns than in non-Finnish–Swedish–Estonian Europeans (derived from gnomAD v3). The ratio of allele frequencies is reported in the Enrichment (FIN) column. All variants except the two marked with asterisks were not present in Pan UKBB results owing to the extreme low frequency. Allele frequency (FIN), nonreference allele frequency in the FinnGen cohort; $\beta$ and s.e., $\beta$ and s.e. from FinnGen GWAS (Methods). The Coding association column denotes variants also appearing in Table 1 at which a coding variant is, or is in high LD with, the lead variant. Genome coordinates reported from human genome assembly GRCh38 (hg38). P values are from the FinnGen–UKBB, inverse-variance weighted meta-analysis (Methods) and are uncorrected for multiple testing.

function. Consistent with this observation, Thr155Met also increases risk of immunodeficiencies in FinnGen ($P < 0.0001$).

**Intersection of hypothyroidism with checkpoint inhibition**
Programmed cell death protein 1 (PD-1) checkpoint inhibition to activate systemic immune responses has rapidly emerged as a critical tool in the cancer therapy arsenal, with considerable effort dedicated

to understanding the underlying beneficial mechanisms, including enhancement of T cell priming and activation and reinvigoration of exhausted intratumoral T cells[22,23]. However, immune-related adverse events (irAEs), particularly new-onset autoimmune diseases such as hypothyroidism, type 1 diabetes, colitis, hepatitis, myocarditis and vitiligo, remain an important clinical challenge in the use of these important drugs. Recent observations[24–27], however, have

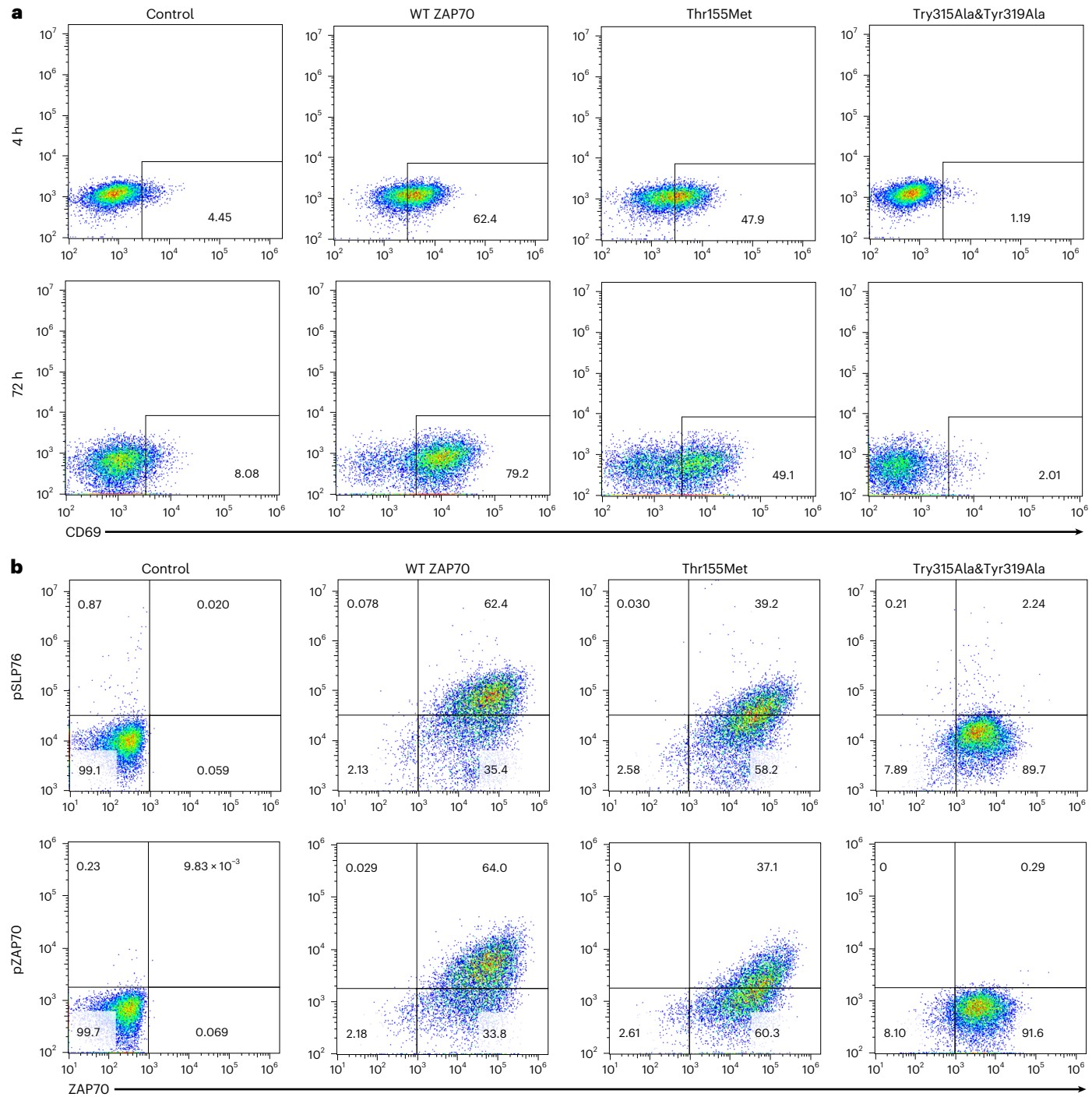

**Fig. 1 | Impaired T cell activation in the ZAP70 Thr155Met variant T cell line in response to TCR stimulation. a,b**, ZAP70-deficient Jurkat cells (P116 clone) reconstituted with either ZAP70 or variants and stimulated with anti-CD3 and anti-CD28 (1 µg ml$^{-1}$). Cells were then stained for detection of CD69 (**a**) or ZAP70, phospho-ZAP70 (pZAP70) and phospho-SLP76 (pSLP76) (**b**) before FACS analysis. Control, ZAP70-deficient cells infected with control virus; Thr155Met, ZAP70-deficient cells expressing ZAP70 Thr155Met; Try315Ala&Tyr319Ala, ZAP70-deficient cells expressing an inactive mutant ZAP70 Try315Ala&Tyr319Ala; WT ZAP70, ZAP70-deficient cells reconstituted with wild-type ZAP70.

demonstrated that individuals with irAEs may be receiving greater benefit from checkpoint inhibition, an observation confirmed in a recent meta-analysis[28]. In one study, anti-PD-L1 atezolizumab-induced thyroid dysfunction was associated with longer survival across seven trials in six cancer types. Furthermore, in one trial, patients with higher hypothyroid polygenic risk scores (PRSs) had both higher rates of atezolizumab-induced thyroid dysfunction and a lower risk of death in triple-negative breast cancer[27].

The relationship between checkpoint inhibition and hypothyroidism in clinical practice, alongside observing individual associations to *CTLA4* and *LAG3*, encouraged us to examine this further. Starting with the other two targets with approved drugs, PD-1 (encoded by *PDCD1*) and its ligand PD-L1 (encoded by *CD274*), we found that the locus at *CD274* contains a nearby upstream genome-wide significant variant (rs911760) (and a second independent signal at neighboring *PDCD1LG1*). Binding of cytotoxic T lymphocyte-associated

protein 4 to CD80 and CD86 prevents continued T cell activation and, among our strongest genome-wide significant associations, we observed variants in LD with the well-described CT60 variant at *CTLA4* (rs3087243), which is correlated with increased cytotoxic T lymphocyte-associated protein 4 levels on CD4$^+$ T cells[29] —consistent with tamping down immunity with consequent lowering of risk to AIHT ($\beta = -0.155$, $P = 2.7 \times 10^{-127}$). We further observed genome-wide significant associations at both *CD80* and *CD86* loci with associations spanning *CD80–TIMMDC1* ($P = 4.3 \times 10^{-14}$) and at the *ILDR1–CD86* locus ($P = 4.4 \times 10^{-10}$). Collectively, these genetic findings suggest a strong relationship between AIHT and the immune checkpoint pathway. The consistency of allelic effects of these AIHT associations with the induced effects of checkpoint immunotherapy supports the idea that irAEs commonly seen in checkpoint immunotherapy represent an on-target effect, as suggested by trial studies[27].

Further to this intersection, we integrated published proteomics data[30], fine-mapped the GWAS signals of 1,500 protein levels and found that 7 AIHT loci were significantly associated with PD-1 levels. Supporting a direct relationship, in all seven, the allele increasing hypothyroid risk increased soluble PD-1 levels, with most in the subgroup that were associated with broader autoimmune disease risk and cancer protection described below (Supplementary Table 2).

## Genetic dissection of hypothyroid risk

**Autoimmunity component.** As AIHT sits at the nexus between thyroid disease and autoimmunity and occurs at a high frequency, we hypothesized that insights into the underpinnings of both systemic autoimmunity and specific thyroid disease processes would be present. To explore this, we performed a similar meta-analysis of individuals with a nonthyroid-based autoimmune disease (Methods) (excluding anyone with any form of thyroid disease). This scan, hereafter termed 'autoimmune nonthyroid (AInonT)' in FinnGen + UKBB, had 70,570 cases and 741,401 controls. Unsurprisingly, there was considerable overlap between AInonT and AIHT scans, with 62 of the 417 index variants from AIHT showing $P < 1.2 \times 10^{-4}$ (0.05/417) and 96 with $P < 0.0024$ (1/417), including 92 of 96 with the same direction of effect (Supplementary Table 3).

**Thyroid-specific component.** Distinct from autoimmunity, congenital hypothyroidism most often results from gene defects in thyroid development (agenesis or dysgenesis) or in thyroid hormone production (dyshormonogenesis) and has been explored in animal and cellular models[31]. Crossreferencing our GWAS signals within 100 kb of genes on the Genomics England clinical panel for congenital hypothyroidism (https://panelapp.genomicsengland.co.uk/panels/31/download/34/) indicates that independent common variation at six of these gene loci are associated with AIHT (Supplementary Table 4), providing pointers to the thyroid-specific effects in our scan.

Expanding to population-wide hormone production variability, thyroid-stimulating hormone (TSH) levels are broadly used in clinical settings to diagnose hypothyroidism. Recent publications[32,33] provided a scan of population-wide TSH levels across multiple biobanks, demonstrating a strong polygenic architecture with hundreds of genome-wide significant associations to serum TSH levels. Using clinical laboratory values available on FinnGen participants since 2014, we performed a serum TSH scan on 226,947 AIHT controls—ensuring independence from the AIHT case–control scan and enabling identification of AIHT associations arising from direct impact on thyroid development or function. Our 417 AIHT index variants similarly show a highly significant excess of overlapping associations with 124 (at $P < 1.2 \times 10^{-4}$) and 152 (at $P < 0.0024$) associated with TSH levels (Supplementary Table 3). Consistent with expectation, 149 of 152 overlaps show that increasing AIHT risk corresponds to higher TSH levels.

To explore shared effects between AIHT and AInonT and TSH in more detail, we applied linemodels, a Bayesian classification algorithm[34], to compare the effect sizes of the 417 AIHT index variants with those in the AInonT and TSH scans separately. We specifically asked whether a model in which there are two groups of variants (roughly 'shared' and 'AIHT specific') fits the observed effect sizes better than a single relationship and, in the two-group case, assigned group membership probabilities to each variant. Comparing AIHT to AInonT, a 2-group solution (termed AIHT only and shared AI) was a vastly better fit with many variants assigned strongly to one or the other group (57 having ≥99% confidence of being shared (termed AIHT–AInonT-99 below) and 71 having ≥99% confidence of being associated with AIHT only) (Fig. 2 and Supplementary Table 3). Running the same comparison between AIHT and TSH summary statistics produced an even more tail-heavy posterior assignment probability distribution between two groups with 51 having >99% confidence in the shared AIHT–TSH group (termed AIHT–TSH-99), whereas there were 231 with 99% confidence in the 'AIHT-only' group.

Notably, these sharing groupings were significantly nonoverlapping: the 57 AIHT–AInonT-99 variants were completely distinct from the 51 AIHT–TSH-99 variants and Spearman's correlation across all 417 AIHT loci between the AIHT–AInonT and AIHT–TSH sharing probabilities was highly significant ($\rho = -0.36$, $P = 2.4 \times 10^{-14}$; Supplementary Table 3). As weaker associations are less able to be assigned confidently to shared or nonshared classes, we estimated sharing proportions from the top half of associations (204 AIHT index variants with $P < 1 \times 10^{-11}$) and observed that 38% of associations are likely shared with AInonT ($P > 0.8$) and 20% shared with TSH (Fig. 2).

Confirming the functional distinction between these sets, we intersected our lead variants with fine-mapping of eQTLs from the expression quantitative trait loci (eQTL) catalog (Supplementary Table 12). Among the 51 AIHT–TSH-99 loci, a strong thyroid excess was seen with 22 mapped to thyroid eQTLs, whereas 4 mapped to T cell eQTLs. By contrast, the 57 AIHT–AInonT-99 loci showed the opposite skew, with 26 mapping to T cell eQTLs and only 3 to thyroid eQTLs. We also utilized the above-mentioned UKBB proteomics data, finding 27 of our index variants significantly associated ($P < 5 \times 10^{-8}$) in *trans* to TSHB (one of two TSH subunits) levels—all with a higher TSHB corresponding to AIHT risk. Of the 27, 26 are among the AIHT–TSH-99 group, with none in the AIHT–AInonT-99 group. Collectively, this confirms that the genetic architecture of AIHT consists of distinct, independent components—one representing processes shared across autoimmune diseases and the other representing thyroid-specific functional contributors.

Autoimmune thyroid disease generally refers to both Hashimoto's disease (hypothyroidism) and Graves' disease (hyperthyroidism). We utilized the FinnGen + UKBB meta-analysis of Graves' disease (Methods), which had a total sample of 6,550 cases and 823,242 controls. Despite the much smaller sample, there was, as expected, a highly significant overlap with AIHT. Among the 417 AIHT index variants were 56 (at $P < 1.2 \times 10^{-4}$) and 101 (at $P < 0.0024$) at the thresholds where 0.05 and 1 are expected by chance. These associations were, however, not unidirectional, with 86 being shared and 15 in the opposite direction. This distinction falls along and reinforces the same dimension described in the earlier linemodels analysis. Specifically, 14 of 15 loci where Hashimoto's disease and Graves' disease have opposite-direction effects are in the AIHT–TSH-99 group (with 0 in AIHT–AInonT-99), whereas, among the 86 shared direction variants, 23 were in the AIHT–AInonT-99 group (with 0 in AIHT–TSH-99). Thus, the two common forms of autoimmune thyroid disease appear to tightly share their autoimmune component, whereas their thyroid-specific component is also shared but acts in opposite directions of risk and protection in the two diseases.

## Inverse genetic risk shared with skin cancer

Although the relationships to autoimmunity and thyroid function are unsurprising, we used the comprehensive phenotypes of FinnGen to explore overlap with other common diseases via both correlation of disease incidence with AIHT genetic risk and by examination of coincident

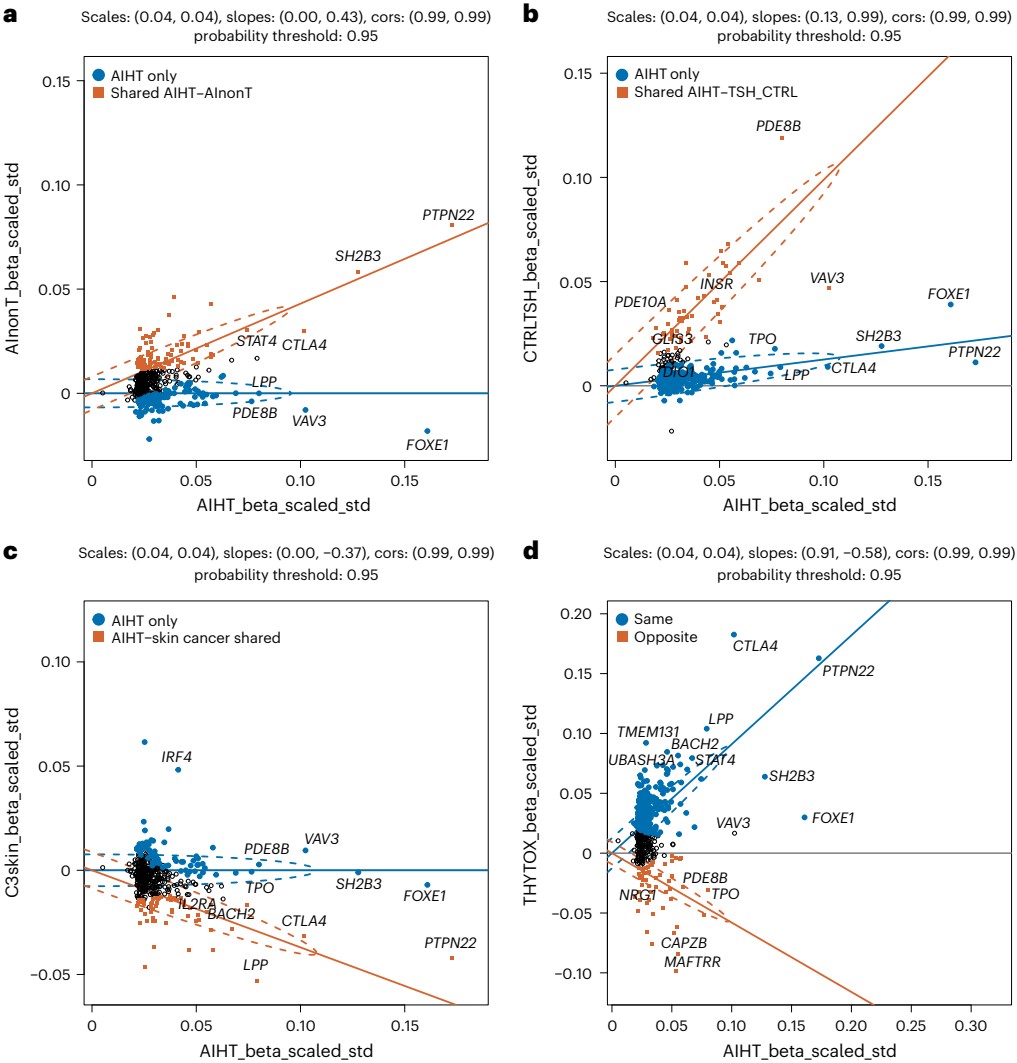

**Fig. 2 | Analysis of shared effects using linemodels. a–d,** Scatter plots of effect sizes from AIHT-associated variant effect sizes compared with effect sizes from nonthyroid autoimmune diseases from FinnGen + UKBB (**a**), TSH levels from FinnGen (**b**), skin cancer from FinnGen + UKBB (**c**) and Graves' disease from FinnGen + UKBB (**d**). Optimal two-group fit of Bayesian linemodels run with scale and correlation parameters were fixed and displayed with red indicating genome-wide significant loci. AIHT PRS was positively correlated (at a conservative threshold of $P < 1 \times 10^{-5}$) with hundreds of FinnGen endpoints. Given the widespread pleiotropy described above, it was unsurprising that there were numerous positive correlations with autoimmune and thyroid disease phenotypes. We also observed significant negative relationships for seven FinnGen cancer endpoints (Supplementary Table 5), led by the incidence of basal cell carcinoma ($r = -0.07$, $P = 1.7 \times 10^{-17}$) and all skin cancers ($r = -0.06$, $P = 8.3 \times 10^{-13}$), with less significant negative overlaps also seen for prostate cancer and the 'all cancer' phenotype.

To explore this further, we performed the FinnGen + UKBB meta-analysis of skin cancer (including melanoma and nonmelanoma: 68,822 cases and 657,740 controls) and examined results at the 417 AIHT index variants (Supplementary Table 3). In this set, there were 13 loci exceeding genome-wide significance, 26 at a comparison-wide level of significance ($P < 1.2 \times 10^{-4}$) and 48 at a level expected once by chance in 417 loci ($P < 0.0024$). Almost as striking as the excess itself, 24 of the 26 (and 42 of the 48) showed effects on risk in skin cancer and hypothyroidism that were in opposite directions.

We used linemodels and identified AIHT-only and AIHT–skin cancer-shared groups, the latter with a negative slope, indicating

an association to AIHT only and blue to both AIHT and the query phenotype. As recommended, $\beta$ and s.e. were transformed into √(heritability) scale by multiplying both by √(2× MAF × (1 − MAF)). Data points are colored when linemodels group assignments are >95% probability. (Full data and assignment probabilities are listed in Supplementary Table 3.) cors, correlations.

variants at which hypothyroid risk alleles correspond to skin cancer protective alleles, and assigned posterior probabilities of assignment to each group for all 417 loci. We then performed Spearman's correlation between the probability membership in the AIHT–skin cancer-shared group with the previously defined probabilities of AIHT–AInonT and AIHT–TSH shared groupings. The AIHT–skin cancer membership was positively correlated with AIHT–AInonT ($\rho = 0.21$, $P = 1.6 \times 10^{-5}$) and negatively correlated with AIHT–TSH sharing ($\rho = -0.24$, $P = 8.1 \times 10^{-7}$), indicating that the shared component of skin cancer and AIHT represents an immune program rather than one specific to the thyroid. Among the 22 variants with >95% posterior assignment to the AInonT–skin cancer shared group are well-known missense variants in *PTPN22*, *TYK2*, *IFIH1*, *FUT2* and *CCDC88B*, as well as the low-frequency *ZAP70* and *IRF3* variants noted above, an intronic *FLT3* variant that prematurely truncates *FLT3* and other established immune-mediated disease loci (*CTLA4*, *BACH2*, *STAT2*, *PTPN2* and *IL2RA*).

To confirm these relationships in independent samples, we examined association of a hypothyroidism PRS calculated from UKBB alone against all phenotypes in FinnGen (Methods). As expected, the PRS

was strongly positively associated with AIHT (odds ratio (OR) = 1.71, $P < 1 \times 10^{-300}$) and numerous other autoimmune diseases, including type 1 diabetes, seropositive rheumatoid arthritis and vitamin B[12] deficiency anemia (all $P < 1 \times 10^{-100}$) (Supplementary Table 6). By contrast, a strong negative association was found between the UKBB PRS and multiple cancer endpoints, led by basal cell carcinoma (OR = 0.91 (95% confidence interval (CI) 0.90–0.92), $P = 3.0 \times 10^{-39}$), all skin cancers (OR = 0.92 (0.91,0.93), $P = 8.4 \times 10^{-36}$) and the umbrella 'all cancer' endpoint (OR = 0.96 (0.95, 0.970), $P = 6.0 \times 10^{-26}$) as well as individually significant breast (OR = 0.96 (0.94, 0.97)) and prostate (OR = 0.94 (0.93, 0.96)) cancer endpoints (Supplementary Table 7). Results were robust to two reanalyses (1) removing the MHC from PRS calculation and (2) removing all FinnGen AIHT cases (to confirm independence from correlations with the diagnosed phenotype self-evidently related to the PRS). Naturally, this second analysis removes significant relationships to thyroid-related phenotypes but leaves the cancer and other autoimmune disease relationships intact (Supplementary Tables 8 and 9).

## Discussion

Using the broad diagnostic and medication information of FinnGen and the UKBB, we present here the largest GWAS to date in autoimmune hypothyroidism. Extensive clinical data available in these biobanks, including prescription medication use, provided a more complete ascertainment of cases, while the diagnostic coverage in each permitted the exclusion of other common thyroid conditions. The resultant analysis included a total of 81,718 cases and yielded a total of 417 independent genome-wide significant variants in addition to the MHC, roughly doubling the numbers found in the largest previous studies[5,6].

Of these 417 associations, 67 (16%) contained coding variants that were, or were in high LD with, the lead variant. As coding variants were twice as often found in low-frequency association credible sets, these provided the most interpretable pointers to new biological insights into AIHT. Among the coding variants likely driving associations were low-frequency coding variants in both *IRF3* and *IRF4*, which, similar to the hypomorphic low-frequency variants in *IFIH1* and *TYK2* also seen here, broaden the set of disease-protective perturbations likely acting through the interferon response. This connection is further supported by a common missense variant at *TLR3*, encoding another interferon-inducing component of antiviral immunity and by a common missense variant in the interferon-inducible *IFITM2*. Another notable association at *PER3* connects circadian regulation to AIHT via two tightly linked missense variants previously linked to morning chronotype and now demonstrated to be protective against AIHT.

We characterized *ZAP70*:Thr155Met, which demonstrates partial loss of function and autoimmunity, whereas homozygosity for complete loss-of-function alleles produces severe combined immunodeficiencies[21]. In addition, complex *ZAP70* genotypes have been associated with autoimmunity. For example, compound heterozygosity for loss-of-function Arg192Trp and gain-of-function Arg360Pro variants caused autoimmunity, but required both alleles to precipitate disease[35]. Collectively, human genetics evidence suggests that ZAP70 function must be optimized within a narrow range; partially impaired activity or enhanced activity elicits autoimmunity[20]. Mouse models have been indispensable in demonstrating this concept and establishing mechanism. Hypomorphic ZAP70 alleles from chemical mutagenesis impaired TCR signaling strength, altering thymocyte development and selecting an autoreactive TCR repertoire with double-stranded DNA antibodies and hyper-immunoglobulin E syndrome[36]. Similarly, a spontaneously arising point mutation of *Zap70* in SKG mice caused partial loss of function and impaired negative selection of autoreactive T cells, resulting in arthritis[37]. Adoptive transfer of naive SKG T cells into immunocompromised recipients was sufficient to induce arthritis, suggesting impaired central and peripheral tolerance[38]. The *ZAP70*:Thr155Met variant associated with AIHT appears to similarly impair TCR signaling, resulting in

loss of tolerance, likely through combined effects of impaired negative selection of autoreactive T cells, lymphopenia-induced homeostatic expansion of pathogenic T cells, defective development or function of regulatory T (T[reg]) cells and resistance to peripheral tolerance mechanisms such as T[reg] cell suppression or anergy induction[39]. These findings have critical therapeutic implications: as *ZAP70*:Thr155Met is a partial loss of function, targeting ZAP70 kinase activity with inhibitors to treat autoimmunity could come with unanticipated consequences. Complete inhibition of ZAP70 may ameliorate T cell-driven autoimmunity at the expense of immunodeficiency, whereas partial inhibition of ZAP70 may exacerbate self-tolerance dysregulation.

The power of this GWAS enables not only detection of significant polygenic overlaps with other phenotypes such as nonthyroid autoimmune diseases and TSH levels (neither of which is individually surprising) but also the determination that these particular overlaps make up distinct, nonoverlapping components of AIHT risk. Using Bayesian linemodels, we estimated that 38% of AIHT associations are shared with autoimmune diseases more broadly and 20% are shared with variation that elevates TSH levels unrelated to immunity. The parallel analysis of Graves' disease and AIHT indicates that, although there is widespread same-direction sharing of the autoimmune components, the thyroid or TSH alleles act in opposite directions, consistent with the hyperthyroid versus hypothyroid character of each disease. The linemodels analysis of this pairing underscores two main components: shared alleles with similar, same-direction effects and shared association with opposite-direction effects, underscoring the importance of distinguishing autoimmune thyroid phenotypes in genetic articulation.

PRS analysis (with external PRS tested in FinnGen) demonstrated a correlation of AIHT PRS to lower risk of skin, as well as breast, prostate and 'all' cancer phenotypes. The 'all cancer' result suggests a broad protective signal shared by most or all cancers, with the sample size of breast and prostate simply being sufficiently large to be individually detected. However, the protective effect on skin cancer was significantly greater, with nonoverlapping CIs compared to all cancers. Considerable sharing between the AIHT and skin cancer GWASs indicated that the opposite-effect alleles conferring risk to AIHT and protection from skin cancer were concentrated in the autoimmune component of AIHT and unrelated to thyroid function specifically. Although skin cancer, and particularly basal cell carcinoma, showed uniquely strong opposite-direction effects, the same highly significant observation in prostate and breast cancer suggests that this is most likely a consequence of general immune surveillance and response to emergent solid tumors, which, although likely of differential relevance to different tumor types, are not specific to skin.

Furthermore, AIHT-associated genetic variants implicate most genes encoding successful targets of checkpoint immunotherapy, including a new rare coding variant in *LAG3*. In addition, seven AIHT risk alleles are significantly associated with soluble PD-1 levels in recently published proteomic data from the UKBB. Moreover, the genetic intersection between AIHT risk and protection from skin cancer sheds light on published observations that individuals with irAEs receive greater benefits from checkpoint immunotherapy. We demonstrated here the same hypothyroid genetic risk that predisposes to thyroid irAEs and improved immunotherapy outcomes also represent a general population-wide signature of cancer protection. This suggests that genetically mediated variation in immune surveillance or function, partially encoded in checkpoint genes, is an important contributor to interindividual variation in cancer risk and potentially highlights mechanisms that could be effective in prevention as well as treatment.

## Online content

Any methods, additional references, Nature Portfolio reporting summaries, source data, extended data, supplementary information, acknowledgements, peer review information; details of author contributions

and competing interests; and statements of data and code availability are available at https://doi.org/10.1038/s41588-026-02521-1.

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

## Methods

### FinnGen ethics statement and study cohort

The FinnGen study (https://www.finngen.fi/en) is a public–private partnership founded in 2017, including Finnish universities, biobanks and hospital districts, as well as several pharmaceutical companies. The aim was to collect both National Health Records and genetic data from 500,000 Finns. The study participants included patients with acute and chronic diseases, healthy volunteers and population collections. R12 consisted of 520,210 individuals (55% women and 45% men, average age 61.9 years). Study participants in FinnGen provided informed consent for biobank research, based on the Finnish Biobank Act. Alternatively, separate research cohorts, collected before the Finnish Biobank Act came into effect (in September 2013) and the start of FinnGen (August 2017), were collected based on study-specific consents and later transferred to the Finnish biobanks after approval by Fimea (Finnish Medicines Agency), the National Supervisory Authority for Welfare and Health. Recruitment protocols followed the biobank protocols approved by Fimea. The Coordinating Ethics Committee of the Hospital District of Helsinki and Uusimaa (HUS) statement no. for the FinnGen study, under which this research is conducted, is HUS/990/2017.

The FinnGen study is approved by the Finnish Institute for Health and Welfare (permit numbers THL/2031/6.02.00/2017, THL/1101/5.05.00/2017, THL/341/6.02.00/2018, THL/2222/6.02.00/2018, THL/283/6.02.00/2019, THL/1721/5.05.00/2019 and THL/1524/5.05.00/2020); digital and population data service agency (permit numbers VRK43431/2017-3, VRK/6909/2018-3 and VRK/4415/2019-3); the Social Insurance Institution (permit numbers KELA 58/522/2017, KELA 131/522/2018, KELA 70/522/2019, KELA 98/522/2019, KELA 134/522/2019, KELA 138/522/2019, KELA 2/522/2020 and KELA 16/522/2020); Findata (permit numbers THL/2364/14.02/2020, THL/4055/14.06.00/2020, THL/3433/14.06.00/2020, THL/4432/14.06/2020, THL/5189/14.06/2020, THL/5894/14.06.00/2020, THL/6619/14.06.00/2020, THL/209/14.06.00/2021, THL/688/14.06.00/2021, THL/1284/14.06.00/2021, THL/1965/14.06.00/2021, THL/5546/14.02.00/2020, THL/2658/14.06.00/2021 and THL/4235/14.06.00/2021); Statistics Finland (permit numbers TK-53-1041-17 and TK/143/07.03.00/2020 (earlier TK-53-90-20), TK/1735/07.03.00/2021 and TK/3112/07.03.00/2021); and Finnish Registry for Kidney Diseases permission or extract from the meeting minutes on 4 July 2019.

The Biobank Access Decisions for FinnGen samples and data utilized in FinnGen Data Freeze 12 include: THL Biobank BB2017_55, BB2017_111, BB2018_19, BB_2018_34, BB_2018_67, BB2018_71, BB2019_7, BB2019_8, BB2019_26, BB2020_1 and BB2021_65; Finnish Red Cross Blood Service Biobank 12 July 2017, Helsinki Biobank HUS/359/2017, HUS/248/2020, HUS/430/2021 §28, §29, HUS/150/2022 §12, §13, §14, §15, §16, §17, §18, §23, §58, §59, and HUS/128/2023 §18; Auria Biobank AB17−5154 and amendment no. 1 (17 August 2020) and amendments BB_2021-0140, BB_2021-0156 (26 August 2021, 2 February 2022), BB_2021-0169, BB_2021-0179, BB_2021-0161, AB20-5926 and amendment no. 1 (23 April 2020) and its modifications (22 September 2021) BB_2022-0262, BB_2022-0256; Biobank Borealis of Northern Finland_2017_1013, 2021_5010, 2021_5010 Amendment, 2021_5018, 2021_5018 Amendment, 2021_5015, 2021_5015 Amendment, 2021_5015 Amendment_2, 2021_5023, 2021_5023 Amendment, 2021_5023 Amendment_2, 2021_5017, 2021_5017 Amendment, 2022_6001, 2022_6001 Amendment, 2022_6006 Amendment, 2022_6006 Amendment, 2022_6006 Amendment_2, BB22-0067, 2022_0262, 2022_0262 Amendment; Biobank of Eastern Finland 1186/2018 and amendment 22§/2020, 53§/2021, 13§/2022, 14§/2022, 15§/2022, 27§/2022, 28§/2022, 29§/2022, 33§/2022, 35§/2022, 36§/2022, 37§/2022, 39§/2022, 7§/2023, 32§/2023, 33§/2023, 34§/2023, 35§/2023, 36§/2023, 37§/2023, 38§/2023, 39§/2023, 40§/2023 and 41§/2023; Finnish Clinical Biobank Tampere MH0004 and amendments (21 February 2020 and 6 October 2020); BB2021-0140 8§/2021, 9§/2021, §9/2022, §10/2022, §12/2022, 13§/2022, §20/2022, §21/2022, §22/2022, §23/2022, 28§/2022, 29§/2022, 30§/2022, 31§/2022, 32§/2022, 38§/2022, 40§/2022, 42§/2022 and 1§/2023; and Central Finland Biobank 1-2017, BB_2021-0161, BB_2021-0169, BB_2021-0179, BB_2021-0170, BB_2022-0256, BB_2022-0262 and BB22-0067. Decision was made allowing continuation of data processing until 31 August 2024 for the following projects: BB_2021-0179, BB22-0067, BB_2022-0262, BB_2021-0170, BB_2021-0164, BB_2021-0161 and BB_2021-0169; and Terveystalo Biobank STB 2018001 and amendment 25 August 2020, Finnish Hematological Registry and Clinical Biobank decision 18 June 2021, Arctic biobank P0844: ARC_2021_1001.

### Phenotypic definitions in FinnGen and UKBB

Exact definitions of International Classification of Diseases (ICD; 8th[40,41], 9th[42,43] and 10th[44,45] edns) diagnoses, medications and procedures for all FinnGen phenotypes are publicly available at https://risteys.finregistry.fi/ using the tag names listed in the summary below. To create a large but specific set of individuals with AIHT, we first collected all individuals with 1+ years of levothyroxine purchases (H03AA01), ICD-10 codes E03[89]x, ICD-9 244[89]X and ICD-8 244[99,00]. From the cases, we then excluded anyone with thyrotoxicosis E05[01289], thyroidectomy (NOMESCO code BAA60), postsurgical hypothyroidism (E89.0[19]), pituitary tumor (D35.2), panhypopituitarism (E23.00), hypopituitarism (E23.08), hypogonadotropic hypogonadism (E32.04), lack of adrenocorticotropic hormone (E23.03) or deficiency of growth hormone (E23.01). Controls were everyone else in FinnGen, excluding those with any of the autoimmune codes in Supplementary Table 11 or listed at https://risteys.finngen.fi/endpoints/AUTOIMMUNE. Detailed descriptions of the FinnGen phenotype can be found at https://risteys.finregistry.fi/endpoints/E4_HYTHY_AI_STRICT.

The UKBB phenotype was created by executing the FinnGen endpoint definition code using the identical ICD codes for inclusion and exclusion. Details specific to UKBB include levothyroxine obtained from 'treatment/medication code' (20003) and 'GP prescription records' (42039) and self-reported hypothyroidism or myxoedema (1226). In addition, to match FinnGen exclusion criteria in the UKBB, thyroidectomy was defined as operation code 1432 or operative procedures B081−B084 (main or secondary OPCS4).

To discriminate autoimmune versus thyroid loci, we created a set of individuals with autoimmune disease but who did not have hyperthyroidism or AIHT. The complete list of selected autoimmune diseases is listed at https://risteys.finregistry.fi/endpoints/AUTOIMMUNE. For the phenotype AUTOIMMUNE_NONTHYROID in the UKBB, from the selected individuals with autoimmune disease, we then excluded those with thyroidectomy, use of levothyroxine or carbimazole, any individuals in the strict AIHT cases described above and those with AIHT (ICD-10: E05[0 | 9] ICD-9: 2420).

The skin cancer endpoint used incorporated all melanoma and nonmelanoma skin cancers and can be viewed at https://risteys.finngen.fi/endpoints/C3_SKIN_EXALLC. These captured all instances of skin cancers recorded in hospital discharge or death registries (ICD-10: C43, C44, ICD-8 or ICD-9: 172−173) and cancer registry (ICD-O-3: C44). The combined FinnGen–UKBB scan has 68,822 cases and 657,740 controls and is available at https://metaresults-ukbb.finngen.fi/pheno/C3_SKIN_EXALLC.

Graves' disease was defined using thyrotoxicosis with diffuse goiter (ICD-10: E05.0, ICD-9: 2420) https://risteys.finngen.fi/endpoints/E4_THYTOXGOITDIF. The combined FinnGen–UKBB scan has 6,550 cases and 823,242 controls and is available at https://metaresults-ukbb.finngen.fi/pheno/E4_THYTOXGOITDIF.

### Meta-analysis and definition of LD-independent associations

Array-based genotype data in FinnGen were called and subjected to variant and sample-level quality control, followed by phasing and imputation (using a panel of 8,554 deeply sequenced Finnish whole genomes) using Eagle 2.3.5 and Beagle 4.1 (described further at https://finngen.gitbook.io/documentation/methods/genotype-imputation/genotype-imputation). In FinnGen DF12, this project-wide process resulted in a

total of 500,348 individuals after removal of related individuals and non-Finnish ancestry people and were used in all FinnGen analyses. FinnGen data analysis pipelines are freely available at https://github.com/FINNGEN/; the FinnGen Handbook, https://finngen.gitbook.io/documentation/, contains a detailed description of data production and analysis, including code used to run analyses. GWAS analysis was performed using REGENIE 2.2.4 and a logistic mixed model adjusted for age, sex, genotyping batch and the first ten principal components (PCs) of ancestry with an approximate Firth test for robust effect size estimation. UKBB analysis was performed using the pipeline implemented by the Pan UKBB project[46] (https://pan.ukbb.broadinstitute.org/) using SAIGE with age, sex, age × sex, age$^2$, age$^2$ × sex and 10 ancestry PCs. Meta-analysis of the FinnGen R12 and UKBB European ancestry subset ($n = 420,531$ after quality control and population clustering described on the Pan UKBB website) was performed using inverse-variance weighted meta-analysis.

As high-resolution fine-mapping algorithms have not been shown to be fully reliable in the context of meta-analyses, particularly when performed using different genotyping and imputation techniques[47], we opted conservatively to flag only index variants representing the most significantly associated variant in confidently LD-independent loci (Supplementary Table 1). Many papers use a fixed threshold such as $r^2 < 0.05$ to define LD independence, but, with numerous associations with $P$ values far below $1 × 10^{-100}$; such a threshold is inadequate and false but apparently genome-wide significant peaks will occur. We defined a stricter definition of LD-independent associations as follows:

- Starting with the most significant association with $P < 5 × 10^{-8}$ on each chromosome, around each genome-wide significant variant, a ±2-Mb window was screened.
- A dynamic LD threshold for each association is defined as $T = \min(0.1, r_s)$, where $r_s$ is defined as the $r^2$ value at which the expected residual $\chi^2$ would be 5.0. This threshold is trivially computed because the expected $\chi^2$ of an LD neighbor is $r^2 × \chi^2$ of the causal variant, so conservatively sets a dynamic threshold that leaves expected signal vastly below genome-wide significance.
- Secondary associations were therefore counted as independent only if they were genome-wide significant and $r^2$ to any more significant association was $<T$. As a result of potential inaccuracy of low values of $r^2$, at particularly strong associations where $T < 0.02$ (that is, residual association signal may exist even at very low values of $r^2$), secondary associations were not defined within 1 Mb of such signals. Nearby signals were confirmed as independent using full conditional analysis in FinnGen on top signals using REGENIE and, owing to the limited accuracy of pairwise LD inference beyond two signals, we only reported two signals within 1 Mb here with the exception of a handful of examples where three signals within 1 Mb were all conditionally independently associated (the third significant after conditioning on the first two simultaneously) at genome-wide significance in FinnGen. Full conditional regional analyses can be browsed at r12.finngen.fi.

## ZAP70 functional studies
ZAP70-deficient Jurkat cells were obtained from American Type Culture Collection. Cells were reconstituted with ZAP70 (wild-type (WT), Thr155Met, Tyr315Ala&Tyr319Ala) and stimulated with anti-CD3 and anti-CD28 (1 µg ml$^{-1}$). Cells were fixed and permeabilized with Cytofix or Cytoperm buffer (BD Biosciences) and stained with the indicated antibodies. Flow cytometry was performed on a CytoFLEX LX (Beckman Coulter) flow cytometer and analysis was performed using FlowJo software.

Cell frequencies in each indicated gate are reported as percentages of total live singlet cells. No sorting was performed. Live cells were selected based on forward scatter (FSC) and/or side scatter (SSC) profiles. Singlets were selected based on FSC-H/FSC-A profiles. Quadrant plots were generated for the indicated antibody markers.

## Bayesian classification of association results
Linemodels (https://github.com/mjpirinen/linemodels) was used to explore the existence of, and classify individual variants into, clusters based on bivariate effects. Using as input the ($\beta$, s.e.) pairing from two GWAS analyses for a set of variants, linemodels probabilistically clusters variants into groups, providing both a likelihood of each number of groups and posterior probability of assignment to each group. Linemodels consists of three parameters: scale (the magnitude of effect), slope (the multiplicative relationship between the effects on each phenotype) and correlation (the expected consistency with the expected values). As recommended, $\beta$ and s.e. were transformed into $\sqrt{}$(heritability) scale by multiplying both by $\sqrt{(2 × MAF × (1 − MAF))}$ before fitting linemodels. The initial value of the scale parameter is set such that 95% of the effect sizes are within twice the scale parameter for all groups. We also chose a correlation parameter of 0.99 to permit modest deviation from the exact best-fit slope. Models with one or two slopes were fit using the EM-algorithm implemented in linemodels and the likelihood ratio test used to compare models. In two-line models, the scale parameters were fixed to be equal. For the comparisons involving AInonT and SKIN, one of the slopes was set to 0 to capture only those variants that belong to AIHT, whereas the other slope was optimized with an EM-algorithm. The slopes for shared groups were 0.43 for AInonT and −0.37 for SKIN. When running linemodels for TSH, both slopes were allowed to be optimized (because AIHT is so common, purely autoimmune associations will induce a residual effect on TSH population wide) and were found to be 0.13 and 0.99. After optimizing slopes, we used an iterative Gibbs Sampler to assign group probabilities.

## PRS PheWAS
We conducted a phenome-wide association study (PheWAS), investigating the associations between a PRS for hypothyroidism, derived from the UKBB data[48], and 4,739 phenotypes from the FinnGen R11. We excluded 149 endpoints with <50 cases from the analysis. We applied logistic regression with *LDLT* decomposition using the 'fastglm' R package (https://CRAN.R-project.org/package=fastglm). The association between the standardized PRS vector and each endpoint was adjusted for sex, age, age$^2$, the first six PCs and genotyping array features.

Our primary focus was the relationship to PRS, with other factors included to address potential residual confounding, but providing no additional meaningful information for our study. Sex-specific traits were run in only the appropriate sex. We set Bonferroni's threshold at $P < 1.05 × 10^{-5}$ (0.05/4,739) in consideration of the multiple tests examined.

In addition to the described experiment, we conducted two modified PheWAS analyses to further explore our findings. The first, an 'exclusion-PRS-PheWAS'[49], aimed to determine whether the secondary trait associations with the hypothyroidism score were influenced by overlapping samples between the hypothyroidism endpoint (E4_HYTHY_AI_STRICT) and studied phenotype. For this, we removed hypothyroidism cases from the analysis and conducted a PheWAS on such filtered cohorts. The second study, a 'noMHC-PRS-PheWAS', examined whether the associations that we discovered were driven by the presence of the MHC locus. In this study, we excluded variants located in the MHC region (6:28510120–33480577; GRCh38) from the PRS model and then assessed the phenome-wide associations using the modified PRS.

## Reporting summary
Further information on research design is available in the Nature Portfolio Reporting Summary linked to this article.

## Data availability

Full GWAS summary statistics are available from the FinnGen public download site: https://www.finngen.fi/en/access_results. All GWAS meta-analysis results utilized in this study are available at metaresults-ukbb.finngen.fi and mvp-ukbb.finngen.fi, with FinnGen results available at r12.finngen.fi and labvalues.finngen.fi and Pan UKBB results available at pan.ukbb.broadinstitute.org. Underlying individual-level data used in this study are available as follows: UKBB data utilized (genetic, phenotypic and proteomic) are available through procedures described at https://www.ukbiobank.ac.uk/enable-your-research. FinnGen as a research project is granted use of national healthcare data and biospecimens according to national and European regulations (GDPR), which preclude the research project from distributing individual-level data. However, any researcher can apply for the health register data from the Finnish Data Authority Findata (https://findata.fi/en/permits/) and for all FinnGen individual-level genotype data (and other profiling data) generated by the project from Finnish biobanks via the Fingenious portal (https://site.fingenious.fi/en/) hosted by the Finnish Biobank Cooperative FINBB (https://finbb.fi/en/). Summary statistics from all FinnGen analyses are available publicly. More details about accessing other FinnGen results can be found at https://www.finngen.fi/en/access_results. Academic users wishing to work with the FinnGen project resource directly can follow the procedures described at https://www.finngen.fi/en/how-we-collaborate.

## Code availability

Central data analysis and processing pipelines used in this project are freely available: fine-mapping pipeline (https://github.com/FINNGEN/finemapping-pipeline); meta-analysis (https://github.com/FINNGEN/META_ANALYSIS); genetic ancestry and PC analysis pipeline (https://github.com/FINNGEN/pca_kinship); GWAS SAIGE pipeline (https://github.com/FINNGEN/saige-pipelines); https://finngen.gitbook.io/documentation/ contains a detailed description of data production and analysis procedures in FinnGen including code used to run analyses; https://github.com/FINNGEN/ contains freely available code repositories used to run analyses in FinnGen. The Bayesian linemodels software is available from https://github.com/mjpirinen/linemodels.

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

## Acknowledgements

N.K. and M.A. were supported by the Aging Biology Foundation (to M.A.). R.J.X. and D.B.G. were supported by the National Institutes of Health (grant nos. DK43351 and DK135492). J. Kero was supported by Sigrid Juselius Foundation Finland (grant no. 1062). S.R. was supported by the Academy of Finland Center of Excellence in Complex Disease Genetics (grant no. 312062). T.T. was supported by the Academy of Finland, University of Helsinki. We thank J. Freudenberg for detailed comments and discussion. We thank A. Pereira, M. Gaziano and the MVP research team for sharing summary statistics.

## Author contributions

M.P.R. designed and led the study and drafted the primary paper. M.P.R., M.K., J. Karjalainen, J.R., M.J.D. and Z.F. performed statistical analyses and interpreted GWAS results. N.K., M.A. and C.A. performed PRS calculations and analyses. D.B.G., R.J.X. and Y.J. carried out the ZAP70 experiments. M.K., K.J.K., T.K., S.L., J.M., M.I.K., T.T. and J. Karjalainen defined phenotypes and performed GWAS analyses. M.J.D., Z.K., M.P., R.J.X., T.T., J.P., J. Kero, M.I.M., A.P. and S.R. provided critical advice, paper editing and other support to the study.

## Funding

## Competing interests

R.J.X. is co-founder of Convergence Bio, Board Director at MoonLake Immunotherapeutics, consultant to Nestlé and a member of Magnet Biomedicine scientific advisory board. M.J.D. is a founder of Maze Therapeutics. M.I.M. and C.A. are employees of Genentech and holders of Roche stock. Z.K. is a former employee of Genentech and holds Roche stock. K.J.K. is on the Nurture Genomics Scientific Advisory Board. The other authors declare no competing interests.

## Additional information

**Correspondence and requests for materials** should be addressed to Mary Pat Reeve.

# Eukaryotic cell lines

Policy information about cell lines and Sex and Gender in Research

| | |
|---|---|
| Cell line source(s) | Zap-70 deficient Jurkat cells, P116 (ATCC ATCC® CRL-2676). |
| Authentication | P116 cells were authenticated by confirmation of Zap-70 deficiency by flow cytometry with Zap-70 specific antibodies. |
| Mycoplasma contamination | P116 cells were confirmed to be mycoplasma negative. |
| Commonly misidentified lines (See ICLAC register) | Not applicable. |

# Plants

| | |
|---|---|
| Seed stocks | *Report on the source of all seed stocks or other plant material used. If applicable, state the seed stock centre and catalogue number. If plant specimens were collected from the field, describe the collection location, date and sampling procedures.* |
| Novel plant genotypes | *Describe the methods by which all novel plant genotypes were produced. This includes those generated by transgenic approaches, gene editing, chemical/radiation-based mutagenesis and hybridization. For transgenic lines, describe the transformation method, the number of independent lines analyzed and the generation upon which experiments were performed. For gene-edited lines, describe the editor used, the endogenous sequence targeted for editing, the targeting guide RNA sequence (if applicable) and how the editor was applied.* |
| Authentication | *Describe any authentication procedures for each seed stock used or novel genotype generated. Describe any experiments used to assess the effect of a mutation and, where applicable, how potential secondary effects (e.g. second site T-DNA insertions, mosiacism, off-target gene editing) were examined.* |

# Flow Cytometry

## Plots

Confirm that:

☒ The axis labels state the marker and fluorochrome used (e.g. CD4-FITC).

☒ The axis scales are clearly visible. Include numbers along axes only for bottom left plot of group (a 'group' is an analysis of identical markers).

☒ All plots are contour plots with outliers or pseudocolor plots.

☒ A numerical value for number of cells or percentage (with statistics) is provided.

## Methodology

| | |
|---|---|
| Sample preparation | Zap70-deficient Jurkat cells were obtained from ATCC. Cells were fixed and permeablized with Cytofix/Cytoperm buffer (BD Biosciences) and stained with the indicated antibodies. |
| Instrument | CytoFLEX LX (Beckman Coulter) flow cytometer. |
| Software | Cytometry analysis was performed with FlowJo software. |
| Cell population abundance | Cell frequencies in each indicated gate are reported as percentages of total live singlet cells. No sorting was performed. |
| Gating strategy | Live cells were selected based on FSC/SSC profiles. Singlets were selected based on FSC-H/FSC-A profiles. Quadrant plots were generated for the indicated antibody markers. |

☒ Tick this box to confirm that a figure exemplifying the gating strategy is provided in the Supplementary Information.

Hematological Registry and Clinical Biobank decision 18th June 2021, Arctic biobank P0844: ARC_2021_1001.

Note that full information on the approval of the study protocol must also be provided in the manuscript.

# Field-specific reporting

Please select the one below that is the best fit for your research. If you are not sure, read the appropriate sections before making your selection.

☒ Life sciences ☐ Behavioural & social sciences ☐ Ecological, evolutionary & environmental sciences

For a reference copy of the document with all sections, see nature.com/documents/nr-reporting-summary-flat.pdf

# Life sciences study design

All studies must disclose on these points even when the disclosure is negative.

| | |
|---|---|
| Sample size | Full sample size of FinnGen and UKBB were used, this sample was more than twice as large as any previous study. |
| Data exclusions | Data and analysis QC as previously published in the flagship paper - https://www.nature.com/articles/s41586-022-05473-8 |
| Replication | Homogeneity of effects between FinnGen and UKBB was used to confirm validity. Further replication was performed during the review process with MVP and is presented in Supplementary Table 1. |
| Randomization | NA |
| Blinding | NA |

# Reporting for specific materials, systems and methods

We require information from authors about some types of materials, experimental systems and methods used in many studies. Here, indicate whether each material, system or method listed is relevant to your study. If you are not sure if a list item applies to your research, read the appropriate section before selecting a response.

## Materials & experimental systems

| n/a | Involved in the study |
|---|---|
| ☐ | ☒ Antibodies |
| ☐ | ☒ Eukaryotic cell lines |
| ☒ | ☐ Palaeontology and archaeology |
| ☒ | ☐ Animals and other organisms |
| ☒ | ☐ Clinical data |
| ☒ | ☐ Dual use research of concern |
| ☒ | ☐ Plants |

## Methods

| n/a | Involved in the study |
|---|---|
| ☒ | ☐ ChIP-seq |
| ☐ | ☒ Flow cytometry |
| ☒ | ☐ MRI-based neuroimaging |

## Antibodies

| | |
|---|---|
| Antibodies used | Phospho-SLP-76 (Tyr128) Monoclonal Antibody (HNDZ55), APC, eBioscience (Thermo Fisher 17-9037-42).

Alexa Fluor® 488 anti-ZAP70 Phospho (Tyr319)/Syk Phospho (Tyr352) Antibody (Biolegend 683711).

PerCP/Cyanine5.5 anti-ZAP70 Phospho (Tyr292) Antibody (Biolegend 693810).

PE-Cy™7 Mouse Anti-Human CD69 (BD 560712).

Ultra-LEAF™ Purified anti-human CD3 Antibody (Biolegend 317326).

Ultra-LEAF™ Purified anti-human CD28 Antibody (Biolegend 302934).

PE anti-human/mouse ZAP-70 Antibody (Biolegend 313403). |
| Validation | All antibodies were commercially sourced with quality control and validation provided by the manufacturer. |

# Data

Policy information about availability of data

All manuscripts must include a data availability statement. This statement should provide the following information, where applicable:

- Accession codes, unique identifiers, or web links for publicly available datasets
- A description of any restrictions on data availability
- For clinical datasets or third party data, please ensure that the statement adheres to our policy

Full GWAS summary statistics for all phenotypes described in the paper will be available for download from a public Google Cloud platform storage bucket upon publication.

# Research involving human participants, their data, or biological material

Policy information about studies with human participants or human data. See also policy information about sex, gender (identity/presentation), and sexual orientation and race, ethnicity and racism.

| Reporting on sex and gender | Sex is used to refer to biological sex and is included as a GWAS covariate. |
| --- | --- |
| Reporting on race, ethnicity, or other socially relevant groupings | NA, FinnGen is a study of Finnish-ancestry individuals. |
| Population characteristics | Population characteristics of the FinnGen study are described in the flagship paper - https://www.nature.com/articles/s41586-022-05473-8 |
| Recruitment | FinnGen involves recruitment from hospital clinics, healthy blood donors, and population epidemiology studies. Recruitment details are further described in the flagship paper - https://www.nature.com/articles/s41586-022-05473-8. |
| Ethics oversight | The FinnGen study (https://www.finngen.fi/en) is a public-private partnership founded in 2017, including Finnish universities, biobanks, and hospital districts, as well as several pharmaceutical companies. The aim is to collect both National Health Records and genetic data from 500,000 Finns. The study participants include patients with acute and chronic diseases, healthy volunteers, and population collections. R12 consists of 520,210 individuals (55% females and 45% males). Study subjects in FinnGen provided informed consent for biobank research, based on the Finnish Biobank Act. Alternatively, separate research cohorts, collected prior the Finnish Biobank Act came into effect (in September 2013) and the start of FinnGen (August 2017), were collected based on study-specific consents and later transferred to the Finnish biobanks after approval by Fimea (Finnish Medicines Agency), the National Supervisory Authority for Welfare and Health. Recruitment protocols followed the biobank protocols approved by Fimea. The Coordinating Ethics Committee of the Hospital District of Helsinki and Uusimaa (HUS) statement number for the FinnGen study, under which this research is conducted, is Nr HUS/990/2017.
The FinnGen study is approved by Finnish Institute for Health and Welfare (permit numbers: THL/2031/6.02.00/2017, THL/1101/5.05.00/2017, THL/341/6.02.00/2018, THL/2222/6.02.00/2018, THL/283/6.02.00/2019, THL/1721/5.05.00/2019 and THL/1524/5.05.00/2020), Digital and population data service agency (permit numbers: VRK43431/2017-3, VRK/6909/2018-3, VRK/4415/2019-3), the Social Insurance Institution (permit numbers: KELA 58/522/2017, KELA 131/522/2018, KELA 70/522/2019, KELA 98/522/2019, KELA 134/522/2019, KELA 138/522/2019, KELA 2/522/2020, KELA 16/522/2020), Findata permit numbers THL/2364/14.02/2020, THL/4055/14.06.00/2020, THL/3433/14.06.00/2020, THL/4432/14.06/2020, THL/5189/14.06/2020, THL/5894/14.06.00/2020, THL/6619/14.06.00/2020, THL/209/14.06.00/2021, THL/688/14.06.00/2021, THL/1284/14.06.00/2021, THL/1965/14.06.00/2021, THL/5546/14.02.00/2020, THL/2658/14.06.00/2021, THL/4235/14.06.00/2021, Statistics Finland (permit numbers: TK-53-1041-17 and TK/143/07.03.00/2020 (earlier TK-53-90-20) TK/1735/07.03.00/2021, TK/3112/07.03.00/2021) and Finnish Registry for Kidney Diseases permission/extract from the meeting minutes on 4th July 2019.
The Biobank Access Decisions for FinnGen samples and data utilized in FinnGen Data Freeze 12 include: THL Biobank BB2017_55, BB2017_111, BB2018_19, BB_2018_34, BB_2018_67, BB2018_71, BB2019_7, BB2019_8, BB2019_26, BB2020_1, BB2021_65, Finnish Red Cross Blood Service Biobank 7.12.2017, Helsinki Biobank HUS/359/2017, HUS/248/2020, HUS/430/2021 §28, §29,  HUS/150/2022 §12, §13, §14, §15, §16, §17, §18, §23, §58, §59, HUS/128/2023 §18, Auria Biobank AB17-5154 and amendment #1 (August 17 2020) and amendments BB_2021-0140, BB_2021-0156 (August 26 2021, Feb 2 2022), BB_2021-0169, BB_2021-0179, BB_2021-0161,  AB20-5926 and amendment #1 (April 23 2020) and it´s modifications (Sep 22 2021), BB_2022-0262, BB_2022-0256, Biobank Borealis of Northern Finland_2017_1013, 2021_5010, 2021_5010 Amendment, 2021_5018, 2021_5018 Amendment, 2021_5015, 2021_5015 Amendment, 2021_5015 Amendment_2, 2021_5023, 2021_5023 Amendment, 2021_5023 Amendment_2, 2021_5017, 2021_5017 Amendment, 2022_6001, 2022_6001 Amendment, 2022_6006 Amendment, 2022_6006 Amendment, 2022_6006 Amendment_2, BB22-0067, 2022_0262, 2022_0262 Amendment, Biobank of Eastern Finland 1186/2018 and amendment 22§/2020, 53§/2021, 13§/2022, 14§/2022, 15§/2022, 27§/2022, 28§/2022, 29§/2022, 33§/2022, 35§/2022, 36§/2022, 37§/2022, 39§/2022, 7§/2023, 32§/2023, 33§/2023, 34§/2023, 35§/2023, 36§/2023, 37§/2023, 38§/2023, 39§/2023, 40§/2023, 41§/2023, Finnish Clinical Biobank Tampere MH0004 and amendments (21.02.2020 & 06.10.2020), BB2021-0140 8§/2021, 9§/2021, §9/2022, §10/2022, §12/2022, 13§/2022, §20/2022, §21/2022, §22/2022, §23/2022, 28§/2022, 29§/2022, 30§/2022, 31§/2022, 32§/2022, 38§/2022, 40§/2022, 42§/2022, 1§/2023, Central Finland Biobank 1-2017, BB_2021-0161, BB_2021-0169, BB_2021-0179, BB_2021-0170, BB_2022-0256, BB_2022-0262, BB22-0067, Decision allowing to continue data processing until 31st Aug 2024 for projects: BB_2021-0179, BB22-0067,BB_2022-0262, BB_2021-0170, BB_2021-0164, BB_2021-0161, and BB_2021-0169, and Terveystalo Biobank STB 2018001 and amendment 25th Aug 2020, Finnish |

# Reporting Summary

## Statistics

For all statistical analyses, confirm that the following items are present in the figure legend, table legend, main text, or Methods section.

| n/a | Confirmed | |
|---|---|---|
| ☐ | ☒ | The exact sample size (*n*) for each experimental group/condition, given as a discrete number and unit of measurement |
| ☒ | ☐ | A statement on whether measurements were taken from distinct samples or whether the same sample was measured repeatedly |
| ☐ | ☒ | The statistical test(s) used AND whether they are one- or two-sided *Only common tests should be described solely by name; describe more complex techniques in the Methods section.* |
| ☐ | ☒ | A description of all covariates tested |
| ☒ | ☐ | A description of any assumptions or corrections, such as tests of normality and adjustment for multiple comparisons |
| ☒ | ☐ | A full description of the statistical parameters including central tendency (e.g. means) or other basic estimates (e.g. regression coefficient) AND variation (e.g. standard deviation) or associated estimates of uncertainty (e.g. confidence intervals) |
| ☐ | ☒ | For null hypothesis testing, the test statistic (e.g. *F*, *t*, *r*) with confidence intervals, effect sizes, degrees of freedom and *P* value noted *Give P values as exact values whenever suitable.* |
| ☐ | ☒ | For Bayesian analysis, information on the choice of priors and Markov chain Monte Carlo settings |
| ☒ | ☐ | For hierarchical and complex designs, identification of the appropriate level for tests and full reporting of outcomes |
| ☐ | ☒ | Estimates of effect sizes (e.g. Cohen's *d*, Pearson's *r*), indicating how they were calculated |

*Our web collection on statistics for biologists contains articles on many of the points above.*

## Software and code

Policy information about availability of computer code

| Data collection | NA |
|---|---|
| Data analysis | Central data analysis and processing pipelines used are freely available: fine-mapping pipeline (https://github.com/FINNGEN/finemapping-pipeline); meta-analysis (https://github.com/FINNGEN/META_ANALYSIS); genetic ancestry and PCA pipeline (https://github.com/FINNGEN/pca_kinship); and GWAS SAIGE pipeline (https://github.com/FINNGEN/saige-pipelines). Please see https://finngen.gitbook.io/documentation/ for a detailed description of data production and analysis including code used to run analyses. Please see https://github.com/FINNGEN/ for further code repositories used to run analyses in FinnGen. GWAS analysis was performed using REGENIE 2.2.4 using a logistic mixed model adjusted for age, sex, genotyping batch, and the first ten principal components of ancestry with an approximate Firth test for robust effect size estimation. Bayesian linemodels calculation - https://github.com/mjpirinen/linemodels PRS logistic regression was computed with LDLT-decomposition using the 'fastglm' R package (https://CRAN.R-project.org/package=fastglm). The association between the standardized PGS vector and each endpoint was adjusted for Sex, Age, Age2, the first six principal components, and genotyping array features. |

For manuscripts utilizing custom algorithms or software that are central to the research but not yet described in published literature, software must be made available to editors and reviewers. We strongly encourage code deposition in a community repository (e.g. GitHub). See the Nature Portfolio guidelines for submitting code & software for further information.

