## [Peer Review File · Nature Genetics]

Genome-wide association analyses of autoimmune hypothyroidism reveal autoimmune and thyroid-specific contributions and an inverse relation with cancer risk

Corresponding Author: Ms Mary Reeve

Version 0:

Decision Letter:

2nd October 2024

Dear Mary Pat,

Your Article "Autoimmune hypothyroidism GWAS reveals independent autoimmune and thyroid-specific contributions and an inverse relation with cancer risk" has been seen by three referees. You will see from their comments below that, while they find your work of interest, they have raised several important points. We are interested in the possibility of publishing your study in Nature Genetics, but we would like to consider your response to these points in the form of a revised manuscript before we make a final decision on publication.

To guide the scope of the revisions, the editors discuss the referee reports in detail within the team, including with the chief editor, with a view to identifying key priorities that should be addressed in revision, and sometimes overruling referee requests that are deemed beyond the scope of the current study. In this case, we ask that you address all technical queries related to the association analyses and their interpretation, providing additional details where needed and placing the findings in context with relevant published studies as requested. We hope you will find this prioritized set of referee points to be useful when revising your study. Please do not hesitate to get in touch if you would like to discuss these issues further.

We therefore invite you to revise your manuscript taking into account all reviewer and editor comments. Please highlight all changes in the manuscript text file. At this stage, we will need you to upload a copy of the manuscript in MS Word .docx or similar editable format.

*2) If you have not done so already please begin to revise your manuscript so that it conforms to our Article format instructions, available

[here](http://www.nature.com/ng/authors/article_types/index.html).

*3) Include a revised version of any required Reporting Summary: <https://www.nature.com/documents/nr-reporting-summary.pdf>

Link Redacted

We hope to receive your revised manuscript within 8-12 weeks. If you cannot send it within this time, please let us know.

Nature Genetics is committed to improving transparency in authorship. As part of our efforts in this direction, we are now requesting that all authors identified as 'corresponding author' on published papers create and link their Open Researcher and Contributor Identifier (ORCID) with their account on the Manuscript Tracking System (MTS), prior to acceptance. ORCID helps the scientific community achieve unambiguous attribution of all scholarly contributions. You can create and link your ORCID from the home page of the MTS by clicking on 'Modify my Springer Nature account'. For more information, please visit www.springernature.com/orcid.

Sincerely,
Kyle

Kyle Vogan, PhD
Senior Editor
Nature Genetics
<https://orcid.org/0000-0001-9565-9665>

Referee expertise:

Referee #1: Genetics, thyroid function

Referee #2: Genetics, thyroid function

Referee #3: Genetics, thyroid function

Reviewers' Comments:

Reviewer #1 (Remarks to the Author):

The authors conducted a GWAS on autoimmune hypothyroidism (AIHT), revealed thyroid-specific genetic contributions, and an inverse relation with cancer risk. A substantial part of the manuscript focusses on the reporting of the association results of missense variants. Unfortunately, I have to admit that there are several issues that require clarification, including the methods addressing the thyroid-specific genetic contributions. Overall, the manuscript would benefit from a more detailed summary of the main findings and its implications. For example, the number of variants belonging to the AIHT or AlnonT groups were reported, but do these variants have something in common, do they belong to specific pathways, etc.? My comments in detail are:

1. In the Abstract, 418 independent associations are stated but in the remaining document 417. Please check. Additionally, I strongly encourage avoiding superlatives like "largest-to-date scan of hypothyroidism" particularly in the fast moving GWAS world, and taking into account the complex and somewhat heterogeneous assessment of hypothyroidism case definition across studies.
2. In this respect, how comparable are the trait definition from Saevarsdottir et al. 2020 and Kichaev et al. 2019 to the current study when assessing the number of newly identified loci? Furthermore, the published hypothyroidism GWAS of Mathieu et al. (doi: 10.1016/j.isci.2022.104992) is missing in the introduction, and should be taken into account when assessing known hits.
3. The results paragraph reporting the relation of MAF and coding variants (line 115): The 5% MAF cutoff seems quite high and somewhat artificial. How do the results change by applying a less common frequency cutoff (i.e. 1%) is used for MAF stratification? Furthermore, the effect direction of the minor allele needs to be taken into account – otherwise, the hypothesized selection effect is not generally plausible. Finally, the statement "higher effect alleles detected in frequency-agnostic GWAS analysis are more often coding than lower effect ones" needs to be justified by references or analyses.
4. The imputation quality of the variants need to be provided. Is the final sample size the same for all variants? LambdaGC or

similar measures to evaluate potential inflation of the p-values are missing. QQ plots and Manhattan plots should be provided to get an overview of the results of the GWAS.

5. The genome build and rs numbers should be added to the GWAS result tables. Additionally, the (nearest) gene names should be added so that these results may be linked by the reader easily to the genes reported in the main text of the manuscript.

6. The methods related to the ZAP70 (T155M) T-cell receptor results are not available. Was there a Supplementary text file missing in the manuscript submission? Btw, the methods for the Graves' disease and skin cancer GWAS meta-analyses are missing, too.

7. Several paragraphs of the Results section "Intersection of hypothyroidism with checkpoint inhibition" seem to be more appropriate in the Discussion and Methods sections. I suggest to conduct a co-localization analyses for testing the association with PD-1 levels instead a single SNP lookup in the proteomics data to distinguish a causal variant from LD associations.

8. The paragraph "Autoimmunity component" has several issues that need clarification. What does the sentence "autoimmune hypothyroidism ... occurs at particularly high frequency" mean in this context (given the 5% disease prevalence)? Which are the 304 variants mentioned in the last sentence, and what is the justification for setting the corresponding p-value threshold at $1/417=0.024$?

9. Please provide more details on how the GWAS results were cross-referencing with the Genomics England clinical panel, i.e. how the GWAS genes were obtained (missense, nearest gene,...?) and what is exactly shown in corresponding Table S4.

10. Regarding the overlap of the AIHT associations with TSH levels, I suggest to use a more recent TSH GWAS, e.g. Sterenborg at al. (PMID: 38291025). The question for using the second $p<0.024$ cutoff applies here, too. Finally, it would be of interest to have a closer look at the significant results that show lower TSH level and an increased AIHT risk.

11. The motivation of the authors for limiting the description of the genetic risk association to the negatively correlated traits, i.e. skin cancer is not quite clear (except that the number of those results is much lower than the positively correlated diseases). Furthermore, this association is not unexpected given the known negative (and partly causal) association between TSH levels and several cancers risks (e.g. by Sterenborg at al., PMID: 38291025). The particular type of cancer that emerges from the current analysis depends also on the power (i.e. number of cases) which is relatively high for skin cancer. Thus, the phrasing "striking" overestimates these results a bit.

12. The analysis of "an association study of a hypothyroidism polygenic risk score (PGS) from UKBB across all phenotypes in FinnGen" is also not quite clear, particularly in which dataset the PGS variants and effects were obtained from, and in which of these two studies they were tested for association.

13. I have several questions regarding the application of the linemodels. Looking at the reference of Pirinen (which should be cited also in the Methods section), it seems not plausible to apply a single scale parameter value for all traits tested, particularly because the effect sizes between the TSH (inverse-normal transformed continuous trait) and AIHT/AlnonT (binary trait using logistic regression) are on different scales. The selection of the scale parameter of 0.6 seems too high (also in comparison with the original methods publication) given that the effect estimates even of the significant AIHT GWAS results are almost all smaller (Table S3). Another motivation of the method is a sample overlap between traits, which is not the case for the TSH GWAS and the current study samples. How valid is this method in this scenario? Finally, the legend and axis description of Figure 2 should be changed to more meaningful terms.

14. Why were age-square and its interaction with sex included only in the UK Biobank GWAS, and are the standard errors of both models comparable? This would be particularly important for the inverse-variance meta-analysis.

15. There are several issues related to the description of the outcome definition. For FinnGen, the diagnosis of hypothyroidism was stated as "most commonly" based on ICD10 E03.9 (Hypothyroidism, unspecified). However, looking at the referenced FinnGen website also additional codes were included, e.g. E03.8 (autoimmune caused hypothyroidism). Please provide a comprehensive list of inclusion (and exclusion) criteria for the case/control definition in the manuscript as this is important for the reader to contextualize the specific trait definition of this complex phenotype.

16. I was not able to find the detailed UKBB phenotype descriptions, i.e. the Supplementary info "Detailed Phenotype Descriptions". Please check if it is really available. Furthermore, I suggest to provide the complete list of selected autoimmune diseases used for the phenotype AUTOIMMUNE_NONTHYROID also in the manuscript (e.g. Supplement) to be independent of the availability modification of the finregistry website.

17. Similar issue applies for the genotype data: please provide a brief summary of the imputation methods (i.e. panels, quality control) applied for the imputation of both datasets which would be helpful for the reader (and reviewer) without having to dig into the references.

18. The method description of the LD-independent associations is a bit hard to follow, particularly the long sentences of the second part. It was not clear to me what "expected residual chi-square" that would be 5.0 refers to. Please elaborate a bit

more on this method. Was the conditional analysis in FinnGen performed only for the variant with the smallest p-value per locus, or also for subsequently identified independent variants? Is this LD-independence approach comparable to the methods used in former hypothyroidism GWAS, particularly when stating the number of newly identified signals?

19. I am sure that there are at least a few limitations of the current study that could/should be stated.

Minor issues:

- Often sentences are very long and hard to follow. Please consider rewriting them.
- Reporting Summary: in the Statistics section, several n/a are selected that rather seem to require a Confirmed. Numbers need to be provided for the Sample size. Please check.
- The GWAS summary statistics will be available for download from a public Google Cloud platform storage bucket upon publication. Is the download free of charge for the scientific community?

Reviewer #2 (Remarks to the Author):

In this study, Reeve and coworkers perform the currently largest GWAS on Hashimoto's hypothyroidism. Strong aspects include the specific phenotype definition, the sample size, and the functional follow-up analyses on ZAP70, while analyses are sound and the paper well-written. However, this reviewer has a number of major concerns:

1. The authors aim to dissect the shared and distinct genetic underpinnings of AITD, non-AITD autoimmune diseases and TSH, concluding that there are distinct groups of genes determining normal range TSH levels and autoimmunity, and that for AITD there is a group of autoimmune and thyroid specific variants. This insight is not new – by simply looking at the GWAS significant hits for thyroid function (TSH, FT4) and AITD in previous GWASs, one directly sees that the former is more driven by non-autoimmune thyroid genes and the latter by both autoimmune and a few non-autoimmune thyroid specific genes. Instead, with the identified novel genes, it would be more interesting to go deeper into the exact immune pathways implicated in AITD and other autoimmune diseases.

2. The authors perform a PheWAS, and as expected associations were detected with many autoimmune diseases. The authors also detected associations with several cancers, eventually concluding that the autoimmune and not the thyroid specific genes are the driving factor behind this. As autoimmune pathways are known to play a role in cancer risk, this finding is not unexpected, while an important part of the results section is dedicated to this including the construction of PRS scores.

Minor remarks:

1. For checking the overlap with genetic basis of TSH levels, the authors use the Zhou GWAS. However, this is not the most recent and largest TSH GWAS, which is Sterenborg et al. Nature Comm 2024.

2. Results L75-81: the description of the phenotype definition is redundant – it is discussed in the methods section, so can be removed from the results. In the rest of this section there is too much emphasis on the fact that a specific phenotype definition is used. One would not expect otherwise (while I agree that previous GWASs have non-specific case definitions), so the description of the analyses including other thyroid diseases should be removed from this section.

3. Results L93: close phenotype analog – this is too vague and cannot be easily found in the methods section. What is the exact phenotype definition in UKBB?

4. Results L133: it is unclear why the authors highlight the PER3 variants and mention their association with sleep patterns, as this has little known relation with autoimmune thyroid disease.

Reviewer #3 (Remarks to the Author):

Summary

The manuscript reports a genome-wide association analysis of autoimmune hypothyroidism (AIHT) in FinnGen (n~55k) and UK Biobank (n~27k). The authors take a careful and nuanced approach to phenotyping which they report provides better power than a broader phenotype with larger sample size. They report 417 independent variants associated at genome-wide significance, of which they state >50% are novel for thyroid disease, although this is not formally analysed as far as I can see and may depend on the comparator. They undertake interesting additional analyses using Bayesian methods which provide evidence for two distinct groups of variants acting through autoimmune and through thyroid-specific mechanisms, and separately demonstrating that shared genetic architecture of AIHT and skin cancer is likely to be autoimmunity-related. They also undertake some functional experiments to show that a missense variant in ZAP70 (T155M) leads to a loss of function which impairs downstream signalling from T-cell receptor activation, potentially underlying its role in immunodeficiency and thyroid disease.

Comments

There are some challenges for this paper as currently drafted. There is no formal replication or validation of the findings. I note that all variants appear to have a consistent direction of effect in FinnGen and UK Biobank, but for a reasonably large number, the significance of the association is largely driven by just one of the cohorts. I appreciate the unique nature of the Finnish population which may pose challenges for replication in some cases, but the same issue doesn't apply to UK Biobank, and some indication of whether novel variant associations can be replicated would be helpful.

In addition, this is a busy area of research currently with much closely related work being undertaken. For example, in July 2024, Figueredo et al published a GWAS of hypothyroidism (along with other thyroid traits) in >58k cases from Estonian Biobank, FinnGen and UK Biobank, which identified 141 variants for hypothyroidism, including the IL21R association. In 2022, Mathieu et al also undertook a similar analysis with slightly smaller discovery sample. The recent GWAS literature on thyroid hormone levels in various cohorts is also little mentioned except for the 2020 paper by Zhou et al, but is also highly relevant.

I also appreciate the thoughtful and nuanced approach which the authors have taken to phenotyping and that this may be a slightly distinct phenotype to previous analyses, but those analyses nevertheless remain very closely related; as the authors demonstrate, a polygenic risk score for simple self-reported hypothyroidism is strongly positively associated with AIHT, and there is substantial overlap with quantitative thyroid function traits. I think it's important to compare variant associations to understand (a) what this GWAS adds and (b) to bring to light, explore and understand differences in findings between different approaches.

The authors employ a number of additional analytical and experimental methods to shed further light on their findings. The Bayesian analysis is an interesting exploration of the likely functional role of the GWAS variants, providing evidence for two distinct groups of variants, one implicated in general autoimmunity, and one specific to thyroid development and/or function. Separately, of 86 variants shared between autoimmune hyperthyroidism (Graves') and hypothyroidism (Hashimoto's), some were autoimmunity-related, with the same direction of effect on risk of both conditions, and some were thyroid-specific, with opposite direction of effect, although the majority of shared loci were not allocated to either group. It is worth noting that there is a more recent and substantially larger GWAS of TSH (Sterenberg et al) which did not include UKB or FinnGen as far as I'm aware, which I imagine would strengthen this analysis, though I realize the authors may not have been aware of this work at the time of commencing their analysis.

Finally, phenome-wide associations were sought in FinnGen for a PRS constructed from the current GWAS (for which the majority of cases were from FinnGen) and for an existing PRS of general hypothyroidism (trained in UK Biobank, Weissbrod et al, 2022). These reproduced the established positive associations of thyroid traits with autoimmune disease. The PheWAS also showed an inverse relationship with various types of cancer, including skin, breast, and prostate cancer, which reinforce and extend previous findings in relation to thyroid, breast and prostate cancer. The authors take this a step further, undertaking an additional skin cancer GWAS and identifying shared variants with opposite directions of effect, and applying the Bayesian approach again to provide evidence that the shared architecture of skin cancer and AIHT is likely to be autoimmune.

Minor comments

1. Abstract – 418 independent associations are quoted in the abstract; this number is 417 through the rest of the paper. Please clarify.

2. Tables – It would be very helpful to the reader to provide rsids in all tables (where possible)

Version 1:

Decision Letter:

21st May 2025

Dear Mary Pat,

Your revised Article "Autoimmune hypothyroidism GWAS reveals independent autoimmune and thyroid-specific contributions and an inverse relation with cancer risk" has been seen by Reviewer #3. (Reviewers #1 and #2 were not able to comment on the revision.) As you will see from the comments below, Reviewer #3 is generally satisfied but has requested clarification of a few outstanding issues. We remain interested in publishing your study in Nature Genetics, but we would like to see your response to these points in the form of a further revision before we make a final decision.

We therefore invite you to further revise your manuscript taking into account these comments. Please highlight all changes in the manuscript text file. At this stage, we will need you to upload a copy of the manuscript in MS Word .docx or similar editable format.

*2) If you have not done so already, please begin to revise your manuscript so that it conforms to our Article format instructions, available

[here](http://www.nature.com/ng/authors/article_types/index.html).

In particular, please reformat the reference list and reference callouts to follow journal style (i.e., create a numbered reference list), present Table 1a and Table 1b as two separate display items (i.e., Table 1 and Table 2), present both tables in the main article file in editable Word format (rather than presenting them in the Excel file), and ensure that all primary display items (Figures and Tables) are sized to fit comfortably onto a single page so they can be accommodated in the final article layout.

*3) Include a revised version of any required Reporting Summary (<https://www.nature.com/documents/nr-reporting-summary.pdf>).

EXTENDED DATA FIGURES

Link Redacted

We hope to receive your revised manuscript within 4-8 weeks. If you cannot send it within this time, please let us know.

Nature Genetics is committed to improving transparency in authorship. As part of our efforts in this direction, we are now requesting that all authors identified as 'corresponding author' on published papers create and link their Open Researcher and Contributor Identifier (ORCID) with their account on the Manuscript Tracking System (MTS), prior to acceptance. ORCID helps the scientific community achieve unambiguous attribution of all scholarly contributions. You can create and link your ORCID from the home page of the MTS by clicking on 'Modify my Springer Nature account'. For more information, please visit www.springernature.com/orcid.

Sincerely,
Kyle

Kyle Vogan, PhD
Senior Editor
Nature Genetics
<https://orcid.org/0000-0001-9565-9665>

Referee expertise:

Referee #3: Genetics, thyroid function

Reviewers' Comments:

Reviewer #3 (Remarks to the Author):

Thanks to the authors for their response.

In relation to my own comments, I have only minor outstanding points to raise:

- 1) It would be helpful to state the threshold for defining replication in MVP where this is described in the text.
- 2) There are updated results in the section on genetic dissection of hypothyroid risk, now taken from a GWAS of TSH levels within the FinnGen cohort. I note that the text describing the results still refers (p7, end of paragraph 3) to comparison with the "earlier study" - is this a typo?
- 3) Additionally, given this change in approach, does the sample overlap between the two GWAS introduce any potential biases in this analysis?

Version 2:

Decision Letter:

Our ref: NG-A65893R1

30th July 2025

Dear Mary Pat,

Thank you for submitting your revised manuscript "Autoimmune hypothyroidism GWAS reveals independent autoimmune and thyroid-specific contributions and an inverse relation with cancer risk" (NG-A65893R1). Based on your responses to Reviewer #3, we will be happy in principle to publish your study in Nature Genetics as an Article pending final revisions to comply with our editorial and formatting guidelines.

We are now performing detailed checks on your paper, and we will send you a checklist detailing our editorial and formatting requirements soon. Please do not upload the final materials and make any revisions until you receive this additional information from us.

Thank you again for your interest in Nature Genetics. Please do not hesitate to contact me if you have any questions.

Sincerely,
Kyle

Kyle Vogan, PhD
Senior Editor
Nature Genetics
<https://orcid.org/0000-0001-9565-9665>

We thank the reviewers for their thorough and constructive evaluation of this manuscript. We have aspired to address all concerns with the content and analyses and have made substantive additions and re-analyses in several areas including updating TSH GWAS results to a much larger and more powerful dataset, updating and expanding the runs of the Bayesian linemodels, addition of a replication cohort and a more extensive computational and transcriptomic annotation of the associations. These and other suggested specific additions and clarifications, which we feel have strengthened the manuscript, are described in more detail point-by-point below.

Reviewer #1 (Remarks to the Author):

The authors conducted a GWAS on autoimmune hypothyroidism (AIHT), revealed thyroid-specific genetic contributions, and an inverse relation with cancer risk. A substantial part of the manuscript focusses on the reporting of the association results of missense variants. Unfortunately, I have to admit that there are several issues that require clarification, including the methods addressing the thyroid-specific genetic contributions. Overall, the manuscript would benefit from a more detailed summary of the main findings and its implications. For example, the number of variants belonging to the AIHT or AlnonT groups were reported, but do these variants have something in common, do they belong to specific pathways, etc.? My comments in detail are:

1. In the Abstract, 418 independent associations are stated but in the remaining document 417. Please check. Additionally, I strongly encourage avoiding superlatives like “largest-to-date scan of hypothyroidism” particularly in the fast moving GWAS world, and taking into account the complex and somewhat heterogeneous assessment of hypothyroidism case definition across studies.

Apologies for the lack of clarity, 418 is correct, but we consider the entire MHC as one association and owing to the massive pleiotropy and unresolved LD, the further annotation and overlap analyses focused on the 417 non-MHC hits. This has been clarified and superfluous superlatives removed.

2. In this respect, how comparable are the trait definition from Saevarsdottir et al. 2020 and Kichaev et al. 2019 to the current study when assessing the number of newly identified loci? Furthermore, the published hypothyroidism GWAS of Mathieu et al. (doi: 10.1016/j.isci.2022.104992) is missing in the introduction, and should be taken into account when assessing known hits.

Thank you for mentioning this phenotype comparability across studies, as this is important to understanding the results of our study. We have added Supplementary Table 10 to directly compare our phenotyping approach with prior studies mentioned by the reviewers.

Previous studies relied primarily on UKBB data and used broad inclusion criteria based on levothyroxine prescription. Our study differs in the following ways:

- Our phenotyping approach removes potentially confounding sources of hypothyroidism. First, we exclude non-autoimmune hypothyroid causes (such as post-surgical hypothyroidism following thyroid cancer treatment). We also separate Graves' disease from Hashimoto's thyroiditis - two conditions that can share risk alleles acting in both concordant and opposing directions. The removal of 9,330 such

confounding cases results in a more homogeneous phenotype, which enabled us to identify additional loci (231 loci with confounding cases removed versus 204 loci with them included).

- Additionally, our study includes 54,752 cases from FinnGen, representing a two-fold increase in sample size over both the number of cases from UKBB and the version of FinnGen used in the Mathieu paper.

While Saevarsdottir et al. did remove confounding cases due to thyroidectomy, thyroid cancer and drug exposures such as amiodarone, they included thyrotoxicosis cases (ICD10 E05). Mathieu et al. used an early FinnGen release prior to the phenotype refinement presented here, and Kichaev et al. used only UKBB data. We have added additional linemodels analyses and text highlighting the importance of separating the Graves' and Hashimoto's cases to clarify the impact of this on the genetic findings.

3. The results paragraph reporting the relation of MAF and coding variants (line 115): The 5% MAF cutoff seems quite high and somewhat artificial. How do the results change by applying a less common frequency cutoff (i.e. 1%) is used for MAF stratification? Furthermore, the effect direction of the minor allele needs to be taken into account – otherwise, the hypothesized selection effect is not generally plausible. Finally, the statement “higher effect alleles detected in frequency-agnostic GWAS analysis are more often coding than lower effect ones” needs to be justified by references or analyses.

With respect to the enrichment of coding variant associations, the excess is strongest in associations with $MAF < .01$ (6 of 15, 40%), 28% (10 of 36) in the range $.01 < MAF < .05$, and as noted only 8.7% of the 366 associations with $MAF > .05$. There are indeed several interesting points regarding this in the recent literature on GWAS studies and we have added some references to FinnGen and GBMI papers which address this recently. Both the systematic scan of 2000+ phenotypes in FinnGen and the smaller but deeper GBMI studies have shown the effect of MAF where effect sizes and % associations that are coding increase with lower MAF and we have added references here. Numerous studies have shown (example of the effect in Figure 1 in <https://pmc.ncbi.nlm.nih.gov/articles/PMC9903716/>) however that the selection is not directional with respect to the phenotype examined - but generally demonstrates both stronger risk and protective variants at low frequencies. It is not proven whether this is a consequence of directional selection on underlying phenotypes or with 'stabilizing selection' (described in <https://www.biorxiv.org/content/10.1101/2024.06.19.599789v1.full>) but we have clarified our description so that there is no implication that natural selection is acting on the phenotype we are studying.

4. The imputation quality of the variants need to be provided. Is the final sample size the same for all variants? LambdaGC or similar measures to evaluate potential inflation of the p-values are missing. QQ plots and Manhattan plots should be provided to get an overview of the results of the GWAS.

Thanks for pointing out this oversight, info scores have been added to Supplementary Table 1 and we have publicly released an interactive browser with Manhattan and QQ plots that facilitates browsing of the entire genome-wide results (<https://metaresults-ukbb.finnngen.fi/>).

5. The genome build and rs numbers should be added to the GWAS result tables. Additionally, the (nearest) gene names should be added so that these results may be linked by the reader easily to the genes reported in the main text of the manuscript.

We have added rsID, info score and nearest gene columns to Supplementary Table 1. We have also endeavored to provide the user with more options for assigning the causal gene in Supplementary Table 12. ST12 provides e/sQTLs whose credible set contains the lead variant (with thyroid and T-cell QTLs gene's highlighted), the PoPS (Finucane et al) ranking score for the nearest gene, as well as careful edits from the coding table.

6. The methods related to the ZAP70 (T155M) T-cell receptor results are not available. Was there a Supplementary text file missing in the manuscript submission? Btw, the methods for the Graves' disease and skin cancer GWAS meta-analyses are missing, too.

Apologies for the oversight, we have now added a methods section regarding the technical details of the ZAP70 functional studies - as well as details on the skin cancer and Graves' disease definitions used.

7. Several paragraphs of the Results section "Intersection of hypothyroidism with checkpoint inhibition" seem to be more appropriate in the Discussion and Methods sections. I suggest to conduct a co-localization analyses for testing the association with PD-1 levels instead a single SNP lookup in the proteomics data to distinguish a causal variant from LD associations.

As formal fine-mapping is not yet reliable on meta-analyses, we have now added colocalization using the CLPP statistic in eCaviar which, where available supports the previous inference. This consistency is expected since we had reported not pointwise lookups, but only those lookups of our index variants that were in the credible sets of the UKBB pQTL results (which we had run through fine-mapping) - such that distant LD would not infiltrate the table.

8. The paragraph "Autoimmunity component" has several issues that need clarification. What does the sentence "autoimmune hypothyroidism ... occurs at particularly high frequency" mean in this context (given the 5% disease prevalence)? Which are the 304 variants mentioned in the last sentence, and what is the justification for setting the corresponding p-value threshold at $1/417=0.024$?

We have removed the 'particularly' - the (in retrospect awkward) phrasing was meant to emphasize that the utility of the analysis derived from the high frequency of AIHT. Indeed the 304 was a typo and has been replaced with 417. Since the GWAS were non-overlapping, the .05 and 1 over 417 refer to thresholds expected in .05 and 1 of the lookups of AIHT lead variants by chance - unsurprisingly exceeded by almost 100x

9. Please provide more details on how the GWAS results were cross-referencing with the Genomics England clinical panel, i.e. how the GWAS genes were obtained (missense, nearest gene,...?) and what is exactly shown in corresponding Table S4.

As we were searching for GWAS hits at Mendelian loci, rather than the Mendelian variants themselves, ST4 reports the strongest association within 100 kb of the gene footprint. As this covered roughly half a percent of the genome, the observation that 6 GWS hits landed in it was more than expected but not dramatically so. This analysis was mostly performed as a means of annotating potential non-coding associations and has been clarified in the text.

10. Regarding the overlap of the AIHT associations with TSH levels, I suggest to use a more recent TSH GWAS, e.g. Sterenborg at al. (PMID: 38291025). The question for using the second $p < 0.024$ cutoff applies here, too. Finally, it would be of interest to have a closer look at the significant results that show lower TSH level and an increased AIHT risk.

Thanks for this suggestion - we accessed the Sterenborg scan and it is of course much better powered, however the summary statistics were reported only for roughly 8M of the >13M sites in the FinnGen-UKBB analysis (and specifically missing 89 of our 417 index variants). Fortunately during this past Summer FinnGen has finally obtained lab values on the entire population and we have utilized a much larger TSH scan of 341K individuals for this analysis. We confirmed the new FinnGen scan was highly correlated with the Sterenborg results (described further below in remarks to reviewer 2).

There are 3 loci that have opposite directions of effect for AIHT and TSH levels -

FG_gene_most_severe	v	AIHT_beta	AIHT_pval	TSH_FG_IRN_beta	TSH_FG_IRN_pval
DIO1	1:53909897:C:A	-0.043	1.32E-14	0.013	7.25103E-08
TGFB2	1:218408167:T:C	-0.039	1.68E-11	0.032	1.78649E-41
RERE	1:8437247:T:A	-0.035	1.66E-09	0.013	6.29158E-08

11. The motivation of the authors for limiting the description of the genetic risk association to the negatively correlated traits, i.e. skin cancer is not quite clear (except that the number of those results is much lower than the positively correlated diseases). Furthermore, this association is not unexpected given the known negative (and partly causal) association between TSH levels and several cancers risks (e.g. by Sterenborg at al., PMID: 38291025). The particular type of cancer that emerges from the current analysis depends also on the power (i.e. number of cases) which is relatively high for skin cancer. Thus, the phrasing “striking” overestimates these results a bit.

We have clarified that we focused on the negatively correlated traits as all the more numerous positive ones documented in ST6 represent the established pleiotropy among autoimmune and/or thyroid disease for which positive genetic correlation has been well established. Indeed we agree with the reviewer that the negative correlation we see likely extends to all solid tumor cancers as the ‘all cancer’ phenotype has a high level of significance and equivalent effect size to what is observed in breast and prostate cancer - which reinforces the reviewer’s correct point here are almost certainly on this chart because of their high sample size and not because of a larger effect. We have clarified this general point in the discussion. Notably however, the skin cancer effect sizes are substantially larger and ORs which we have added here do not overlap with the general cancer signals. Indeed the Sterenborg paper which we now incorporate, and previous TSH studies, have noted the clear relationship to thyroid cancer (and this recent paper shows but does not extensively remark on a modest breast cancer signal). The signal observed here seems somewhat

distinct - the cancer protection signal we describe is driven by autoimmune rather than TSH-shared loci (and in fact does not influence thyroid cancer risk). We have further clarified the discussion on these points by adding “The effect on all cancers was indistinguishable from breast and prostate cancer, suggesting a broad protective signal that is shared by most/all cancers, the sample size of breast and prostate simply being sufficiently large to be individually detected. Notably, however protective effect on skin cancer however was significantly greater with non-overlapping confidence intervals than seen in all cancers.”

12. The analysis of “an association study of a hypothyroidism polygenic risk score (PGS) from UKBB across all phenotypes in FinnGen” is also not quite clear, particularly in which dataset the PGS variants and effects were obtained from, and in which of these two studies they were tested for association.

We have clarified in the manuscript - for independence, we take a hypothyroid PRS generated from UKBB test statistics, and then report (ST6) the impact on all FinnGen phenotypes. To test this for robustness to the effect of phenotype and consequent treatment, ST 7-9 include additional tests in which AIHT cases in FinnGen were removed, and the MHC was eliminated from calculation owing to its strength and pleiotropy.

13. I have several questions regarding the application of the linemodels. Looking at the reference of Pirinen (which should be cited also in the Methods section), it seems not plausible to apply a single scale parameter value for all traits tested, particularly because the effect sizes between the TSH (inverse-normal transformed continuous trait) and AIHT/AlnonT (binary trait using logistic regression) are on different scales. The selection of the scale parameter of 0.6 seems too high (also in comparison with the original methods publication) given that the effect estimates even of the significant AIHT GWAS results are almost all smaller (Table S3). Another motivation of the method is a sample overlap between traits, which is not the case for the TSH GWAS and the current study samples. How valid is this method in this scenario? Finally, the legend and axis description of Figure 2 should be changed to more meaningful terms.

We have utilized an updated version of linemodels and expanded the description in the methods. Indeed the scale parameter was misreported previously as we generally set an initial value at $\frac{1}{2}$ of 95% of effect size. At the recommendation of Dr. Pirinen, in these runs, we transformed beta and SE into $\sqrt{\text{heritability}}$ scale by multiplying both by $\sqrt{2 \cdot \text{maf} \cdot (1 - \text{maf})}$ such that rare alleles with large SEs did not exceptionally affect the fit. We opted to keep the scales within and across models fixed for consistency in class assignment relative to distance from expectation - though after transformation the optimal scales if allowed to vary were always quite close to the fixed value reported. While it is an advantage of the method that it can manage overlapping samples, the model is quite valid for comparing non-overlapping summary statistics. If there is no overlap between the studies, then we set $r.lkhood = 0$ and the effect size estimates are treated as independent. This introduces no problem for the model.

14. Why were age-square and its interaction with sex included only in the UK Biobank GWAS, and are the standard errors of both models comparable? This would be particularly important for the inverse-variance meta-analysis.

Good question - the precise covariates used in different biobank studies are not identical based on ascertainment, population, and perhaps preference of the research teams. We examined the distribution of

s.e. In this specific case and found that the distribution of standard errors are consistent and in the expected theoretical ratio given the larger case:control ratio in FinnGen.

15. There are several issues related to the description of the outcome definition. For FinnGen, the diagnosis of hypothyroidism was stated as “most commonly” based on ICD10 E03.9 (Hypothyroidism, unspecified). However, looking at the referenced FinnGen website also additional codes were included, e.g. E03.8 (autoimmune caused hypothyroidism). Please provide a comprehensive list of inclusion (and exclusion) criteria for the case/control definition in the manuscript as this is important for the reader to contextualize the specific trait definition of this complex phenotype.

We have added to the Methods more detailed medical coding notes:

“Hypothyroid cases were included those individuals with 1+ years of levothyroxine purchases (H03AA01), ICD10 codes E03[89]x, ICD9 244[89]X, ICDi 244[99,00]. From the cases were then excluded anyone with thyrotoxicosis E05[01289], thyroidectomy (NOMESCO code BAA60), postsurgical hypothyroidism (E89.0[19]), pituitary tumor (D35.2), panhypopituitarism (E23.00), hypopituitarism (E23.08), hypogonadotropic hypogonadism (E32.04), lack of ACTH (E23.03), deficiency of growth hormone (E23.01). Controls were everyone else in FinnGen excluding those with any of the autoimmune codes in Supplementary Table 11 or listed at <https://risteys.finnngen.fi/endpoints/AUTOIMMUNE>.”

16. I was not able to find the detailed UKBB phenotype descriptions, i.e. the Supplementary info “Detailed Phenotype Descriptions”. Please check if it is really available. Furthermore, I suggest to provide the complete list of selected autoimmune diseases used for the phenotype AUTOIMMUNE_NONTHYROID also in the manuscript (e.g. Supplement) to be independent of the availability modification of the finregistry website.

Thank you for noticing this. The UKBB phenotype descriptions have also been added to the Methods. In Supplementary table 11 we have provided all the AUTOIMMUNE_NONTHYROID phenotypes and their ICD10 codes.

17. Similar issue applies for the genotype data: please provide a brief summary of the imputation methods (i.e. panels, quality control) applied for the imputation of both datasets which would be helpful for the reader (and reviewer) without having to dig into the references.

We have added more specific details to the imputation and GWAS analysis methods - imputation was not done specifically by this project and in the case of the UKBB, was distributed by the UK Biobank so is consistent with all other uses of the resource.

18. The method description of the LD-independent associations is a bit hard to follow, particularly the long sentences of the second part. It was not clear to me what “expected residual chi-square” that would be 5.0 refers to. Please elaborate a bit more on this method. Was the conditional analysis in FinnGen performed only for the variant with the smallest p-value per locus, or also for subsequently identified independent variants? Is this LD-independence approach comparable to the methods used in former hypothyroidism GWAS, particularly when stating the number of newly identified signals?

Thanks for highlighting this, we have expanded the algorithm explanation for clarity. We had noted that several recent papers had supplementary tables of associations with large numbers of false 'secondary' signals because of the use of a fixed threshold of $r^2 < 0.05$ in scenarios such as this where there are now a number of associations with $p < 1e-200$ and beyond - where variants with $r^2 < .05$ can easily still exceed genomewide significance. Briefly, we used instead a dynamic threshold corresponding to $5/\chi^2$ of the lead variant - based on the simple relationship that the expected chi-square of an LD neighbor is the chi-square of the causal variant times r^2 between the two sites. Conditional analysis in FinnGen is performed stepwise with each successive SNP added conditioned on all previous more significant variants accepted. Using the browser (e.g., https://r12.finnngen.fi/region/E4_HYTHY_AI_STRICT/1:19242749-19642749) an interface is available that enables review of the region conditional on 1, 2, and 3 loci and to see separate colocalizations with each of those.

19. I am sure that there are at least a few limitations of the current study that could/should be stated.

Minor issues:

- Often sentences are very long and hard to follow. Please consider rewriting them.
- Reporting Summary: in the Statistics section, several n/a are selected that rather seem to require a Confirmed. Numbers need to be provided for the Sample size. Please check.
- The GWAS summary statistics will be available for download from a public Google Cloud platform storage bucket upon publication. Is the download free of charge for the scientific community?

All GWAS summary statistics presented here can be browsed at metaresults-ukbb.finnngen.fi (FinnGen-UKBB meta-analysis) r12.finnngen.fi (FinnGen only) and are available for download free of charge at https://www.finnngen.fi/en/access_results

Reviewer #2 (Remarks to the Author):

In this study, Reeve and coworkers perform the currently largest GWAS on Hashimoto's hypothyroidism. Strong aspects include the specific phenotype definition, the sample size, and the functional follow-up analyses on ZAP70, while analyses are sound and the paper well-written. However, this reviewer has a number of major concerns:

1. The authors aim to dissect the shared and distinct genetic underpinnings of AITD, non-AITD autoimmune diseases and TSH, concluding that there are distinct groups of genes determining normal range TSH levels and autoimmunity, and that for AITD there is a group of autoimmune and thyroid specific variants. This insight is not new – by simply looking at the GWAS significant hits for thyroid function (TSH, FT4) and AITD in previous GWASs, one directly sees that the former is more driven by non-autoimmune thyroid genes and the latter by both autoimmune and a few non-autoimmune thyroid specific genes. Instead, with the identified novel genes, it would be more interesting to go deeper into the exact immune pathways implicated in AITD and other autoimmune diseases.

2. The authors perform a PheWAS, and as expected associations were detected with many autoimmune diseases. The authors also detected associations with several cancers, eventually concluding that the autoimmune and not the thyroid specific genes are the driving factor behind this. As autoimmune pathways are known to play a role in cancer risk, this finding is not unexpected, while an important part of the results section is dedicated to this including the construction of PRS scores.

Minor remarks:

1. For checking the overlap with genetic basis of TSH levels, the authors use the Zhou GWAS. However, this is not the most recent and largest TSH GWAS, which is Sterenberg et al. Nature Comm 2024.

Indeed we had missed this being published as we completed our analyses but this is certainly a good point. We accessed the Sterenberg scan and it is of course much better powered, however the summary statistics were reported only for roughly 8M of the >13M sites in the FinnGen-UKBB analysis (and specifically missing 89 of our 417 index variants). Fortunately during this past Summer FinnGen has finally obtained lab values on the entire population and we have utilized a much larger TSH scan of 341K individuals for this analysis. To ensure that these TSH scans were interchangeable, for the subset of AIHT index variants present in Sterenberg summary statistics, the correlation between those effect sizes and the ones from the FinnGen study were compared below and demonstrated to be nearly perfectly correlated.

2. Results: L75-81: the description of the phenotype definition is redundant – it is discussed in the methods section, so can be removed from the results. In the rest of this section there is too much emphasis on the fact that a specific phenotype definition is used. One would not expect otherwise (while I agree that previous GWASs have non-specific case definitions), so the description of the analyses including other thyroid diseases should be removed from this section.

We included the before and after phenotypic cleanup to highlight the importance of distinguishing (in particular) forms of hypothyroid that are likely partially or completely distinct in etiology - as it seemed important to R1 and R3 to highlight how this study definitions differ from previous studies, we have retained this in this draft, though agree that the point can now be clearly made with the linemodels analysis comparing autoimmune hypo and hyperthyroid and if agreeable to all, this analysis mapping onto prior studies definition could be moved to supplement.

3. Results L93: close phenotype analog – this is too vague and cannot be easily found in the methods section. What is the exact phenotype definition in UKBB?

Apologies, this was too informally described - we have clarified in the additional methods text:

The UKBB phenotype was created by executing the FinnGen endpoint definition code using the identical ICD codes for inclusion and exclusion. Inclusion in the UKBB, utilized medications used were amiodarone, carbimazole and levothyroxine obtained from “treatment/medication code” (20003) and “GP Prescription records” (42039) and self-reported hypothyroidism/myxoedema (1226). To match FinnGen exclusion criteria in UKBB, thyroidectomy is defined as operation code 1432 or operative procedures B081-B084 (main or secondary OPCS4).

4. Results L133: it is unclear why the authors highlight the PER3 variants and mention their association with sleep patterns, as this has little known relation with autoimmune thyroid disease.

Indeed we highlighted this result because of its unexpected pleiotropic nature - this has been clarified and contextualized with another similarly unexpected observation.

Reviewer #3 (Remarks to the Author):

Summary

The manuscript reports a genome-wide association analysis of autoimmune hypothyroidism (AIHT) in FinnGen (n~55k) and UK Biobank (n~27k). The authors take a careful and nuanced approach to phenotyping which they report provides better power than a broader phenotype with larger sample size. They report 417 independent variants associated at genome-wide significance, of which they state >50% are novel for thyroid disease, although this is not formally analysed as far as I can see and may depend on the comparator. They undertake interesting additional analyses using Bayesian methods which provide evidence for two distinct groups of variants acting through autoimmune and through thyroid-specific mechanisms, and separately demonstrating that shared genetic architecture of AIHT and skin cancer is likely to be autoimmunity-related. They also undertake some functional experiments to show that a missense variant in ZAP70 (T155M) leads to

a loss of function which impairs downstream signalling from T-cell receptor activation, potentially underlying its role in immunodeficiency and thyroid disease.

Comments

There are some challenges for this paper as currently drafted. There is no formal replication or validation of the findings. I note that all variants appear to have a consistent direction of effect in FinnGen and UK Biobank, but for a reasonably large number, the significance of the association is largely driven by just one of the cohorts. I appreciate the unique nature of the Finnish population which may pose challenges for replication in some cases, but the same issue doesn't apply to UK Biobank, and some indication of whether novel variant associations can be replicated would be helpful.

We have now added replication with the Million Veterans Program (MVP) to Supplementary Table 1. Excluding the X chromosome which was not released by MVP, more than 98% of the associations had betas in the same direction as FinnGen-UKBB and 89% achieved nominally significant replication. These meta-analyses have now been released at <https://mvp-ukbb.finnngen.fi>.

In addition, this is a busy area of research currently with much closely related work being undertaken. For example, in July 2024, Figueredo et al published a GWAS of hypothyroidism (along with other thyroid traits) in >58k cases from Estonian Biobank, FinnGen and UK Biobank, which identified 141 variants for hypothyroidism, including the IL21R association. In 2022, Mathieu et al also undertook a similar analysis with slightly smaller discovery sample. The recent GWAS literature on thyroid hormone levels in various cohorts is also little mentioned except for the 2020 paper by Zhou et al, but is also highly relevant.

Indeed we agree and have now added references to the newer publications that emerged during final stages of this study and, as noted above (and relevant to the comments below), have updated to incorporate analysis of the more recent results for TSH.

I also appreciate the thoughtful and nuanced approach which the authors have taken to phenotyping and that this may be a slightly distinct phenotype to previous analyses, but those analyses nevertheless remain very closely related; as the authors demonstrate, a polygenic risk score for simple self-reported hypothyroidism is strongly positively associated with AIHT, and there is substantial overlap with quantitative thyroid function traits. I think it's important to compare variant associations to understand (a) what this GWAS adds and (b) to bring to light, explore and understand differences in findings between different approaches.

We agree - and have expanded the supplementary table 3 to include Graves' disease and more recent TSH study results to highlight the distinctions that are noted when Hashimoto's and Graves are separate rather than combined. In particular, we have added a fourth linemodel comparing Graves' and Hashimoto's, demonstrating two major but opposed sets of associations, one set of alleles with identical risk and one with very strong opposite direction effects - clearly indicating that previous customary treatment of autoimmune thyroid disease as a single entity was not articulating the genetic architecture fully.

The authors employ a number of additional analytical and experimental methods to shed further light on their findings. The Bayesian analysis is an interesting exploration of the likely functional role of the GWAS variants,

providing evidence for two distinct groups of variants, one implicated in general autoimmunity, and one specific to thyroid development and/or function. Separately, of 86 variants shared between autoimmune hyperthyroidism (Graves') and hypothyroidism (Hashimoto's), some were autoimmunity-related, with the same direction of effect on risk of both conditions, and some were thyroid-specific, with opposite direction of effect, although the majority of shared loci were not allocated to either group. It is worth noting that there is a more recent and substantially larger GWAS of TSH (Sternberg et al) which did not include UKB or FinnGen as far as I'm aware, which I imagine would strengthen this analysis, though I realize the authors may not have been aware of this work at the time of commencing their analysis.

Indeed - we were not aware of this when finalizing analyses in our original submission but have now replaced the existing TSH overlap analyses as described in response to R2 in detail.

Finally, phenome-wide associations were sought in FinnGen for a PRS constructed from the current GWAS (for which the majority of cases were from FinnGen) and for an existing PRS of general hypothyroidism (trained in UK Biobank, Weissbrod et al, 2022). These reproduced the established positive associations of thyroid traits with autoimmune disease. The PheWAS also showed an inverse relationship with various types of cancer, including skin, breast, and prostate cancer, which reinforce and extend previous findings in relation to thyroid, breast and prostate cancer. The authors take this a step further, undertaking an additional skin cancer GWAS and identifying shared variants with opposite directions of effect, and applying the Bayesian approach again to provide evidence that the shared architecture of skin cancer and AIHT is likely to be autoimmune.

Minor comments

1. Abstract – 418 independent associations are quoted in the abstract; this number is 417 through the rest of the paper. Please clarify.

Apologies for the lack of clarity, 418 is correct treating the entire MHC as one association. Owing to the massive pleiotropy and unresolved LD in MHC, the further annotation and overlap analyses focused on the 417 non-MHC hits. This has been more clearly stated.

2. Tables – It would be very helpful to the reader to provide rsids in all tables (where possible)

We have added rsID as well as imputation info score and replication to the main table of hits in Supplementary Table 1.

This email has been sent through the Springer Nature Tracking System NY-610A-NPG&MTS

Reviewer #3 (Remarks to the Author):

In relation to my own comments, I have only minor outstanding points to raise:

Many thanks to Reviewer 3 for the continued helpful evaluation of our manuscript.

1) It would be helpful to state the threshold for defining replication in MVP where this is described in the text.

This has been added to the text (the percentages reported were at a one-sided replication of $p < .05$) and full p-value distribution is presented in ST1.

2) There are updated results in the section on genetic dissection of hypothyroid risk, now taken from a GWAS of TSH levels within the FinnGen cohort. I note that the text describing the results still refers (p7, end of paragraph 3) to comparison with the "earlier study" - is this a typo?

Yes indeed - thanks for catching this. This has been corrected and the section further updated per the comment below.

3) Additionally, given this change in approach, does the sample overlap between the two GWAS introduce any potential biases in this analysis?

This is a good question, and one that is likely to occur to readers. To ensure independence from the case-control AIHT analysis, we have altered this analysis to reflect TSH levels in the AIHT control group only - eliminating any background correlation between the analyses (since the association of a quantitative trait within controls is completely statistically independent of the binary case vs. control analysis). In addition to being mathematically preferable, upon reflection, this is also a more sensible way to run the study, since the treatment of AIHT cases would to some extent, confound the genetic relationship between genetic variation and TSH levels so cases would likely contribute less to this scan. ST3 and all derived analyses have been updated with this independent controls-only TSH scan.

The control-only TSH scan shows effect sizes highly correlated to those observed in Sterenberg et al (3rd plot below) for the 328 of 417 loci present in the summary statistics of that study ($r=0.963$, slope=1.046). The results using this updated scan slightly improve the proportion of variants overlapping between the AIHT and TSH studies, and make no substantive changes to the conclusions of the linemodels results that have been updated to this version. As before, there is no overlap between the highest confidence shared variants with TSH and those shared with non-thyroid autoimmunity. All reported Spearman's correlations have been recalculated and are slightly more significant than before.

All supplementary tables have been updated to this improved, independent TSH scan and we thank the reviewer for raising this important issue.